# RNA tertiary structure and conformational dynamics revealed by BASH MaP

Maxim Oleynikov, Samie R Jaffrey*

Department of Pharmacology, Weill Medical College, Cornell University, New York, United States

## eLife Assessment

This **important** work substantially advances our understanding of RNA structure analysis by introducing an innovative method that extends DMS probing to include guanosine residues, thereby enhancing our ability to detect complex tertiary interactions. The evidence supporting the conclusions is **compelling**, with detailed analyses demonstrating the method's capacity to differentiate structural contexts and improve RNA structure predictions. This work will be of broad interest to RNA structural biology, biochemistry, and biophysics researchers.

*For correspondence:
srj2003@med.cornell.edu

**Abstract** The functional effects of an RNA can arise from complex three-dimensional folds known as tertiary structures. However, predicting the tertiary structure of an RNA and whether an RNA adopts distinct tertiary conformations remains challenging. To address this, we developed BASH MaP, a single-molecule dimethyl sulfate (DMS) footprinting method and DAGGER, a computational pipeline, to identify alternative tertiary structures adopted by different molecules of RNA. BASH MaP utilizes potassium borohydride to reveal the chemical accessibility of the N7 position of guanosine, a key mediator of tertiary structures. We used BASH MaP to identify diverse conformational states and dynamics of RNA G-quadruplexes, an important RNA tertiary motif, in vitro and in cells. BASH MaP and DAGGER analysis of the fluorogenic aptamer Spinach reveals that it adopts alternative tertiary conformations which determine its fluorescence states. BASH MaP thus provides an approach for structural analysis of RNA by revealing previously undetectable tertiary structures.

## Introduction

Chemical probing reagents can reveal extensive information about RNA structure. Typically, folded RNAs are incubated with chemical probing agents that modify accessible nucleotides. Atoms that are involved in hydrogen bonding are protected from modification, while unbonded surface accessible atoms are modified. The chemical probing can also be performed with RNA that is systematically mutagenized to predict the role of each nucleotide in the overall RNA structure. In principle, the pattern of modifications can reveal the diverse set of hydrogen-bonding interactions within an RNA, allowing RNA structures to be predicted.

However, not all modifications that are introduced by a chemical probing agent into RNA can be detected. For example, one of the most widely used RNA structure-probing reagents, dimethyl sulfate (DMS), primarily forms three methyl modifications on mRNA: $N^1$-methyladenosine (m1A), $N^3$-methylcytidine (m3C), $N^7$-methylguanosine (m7G) (*Lawley and Brookes, 1963*; *Mustoe et al., 2019*). Among these, m1A and m3C contain methyl modifications located on the Watson-Crick face and therefore can be readily detected due to their ability to impair reverse transcription and thus induce

misincorporations in cDNA (*Peattie and Gilbert, 1980*; *Inoue and Cech, 1985*; *Lempereur et al., 1985*; *Wells et al., 2000*). In contrast, m$^7$G has minimal effects on reverse transcription. Since m$^7$G contains a methyl modification on the Hoogsteen face, rather than the Watson-Crick face of guanosine, it does not efficiently perturb reverse transcription. As a result, the accessibility of the N7 position of G in RNA cannot be readily assessed.

The ability to detect methyl modifications on the Watson-Crick face allows structure-probing methods to accurately predict RNA secondary structure. RNA secondary structure is mediated by Watson-Crick base pairs and includes the location of helices and loop regions in an RNA structure. A's and C's are protected from methylation when they are engaged in Watson-Crick base pairs, and are readily methylated when they are in single-stranded regions. The resulting m$^1$A and m$^3$C residues are the primary nucleotides detected in DMS-based RNA structure mapping methods (*Homan et al., 2014*; *Siegfried et al., 2014*; *Zubradt et al., 2017*).

In contrast to the Watson-Crick face, interactions that occur with the Hoogsteen face have important roles in mediating RNA tertiary structures. RNA tertiary structure includes long-distance interactions that coordinate the overall topology of an RNA as well as other structurally complex non-helical RNA structures. These include G-quadruplexes, which rely on hydrogen bonding interactions of the N7 position of G (N7G) (*Gellert et al., 1962*). G-quadruplexes comprise stacks of two or more G-tetrads, which are square planar arrangements of four guanine bases that interact through hydrogen bonds around a central potassium ion (*Gellert et al., 1962*). G-quadruplexes are physiologically important RNA structures associated with translation, splicing, phase separation, and small molecule binding (*Roschdi et al., 2022*; *Marcel et al., 2011*; *Blice-Baum and Mihailescu, 2014*; *Zhang et al., 2019b*; *Asamitsu et al., 2023*; *Rodriguez et al., 2008*). G-quadruplex folding also appears to be dynamically regulated by specific helicases (*Creacy et al., 2008*; *Guo and Bartel, 2016*). However, the inability to probe the accessibility of N7G in RNA has limited DMS structure mapping experiments from resolving RNA structures such as G-quadruplexes which depend on N7G interactions.

Here. we describe BASH MaP (Borohydride Assisted Structure determination of N7G Hoogsteen interactions through Misincorporation Profiling), a method for determining RNA tertiary structure by measuring global patterns of N7G accessibility in RNA. BASH MaP detects DMS-induced modifications on both the Watson-Crick and Hoogsteen faces by using potassium borohydride to convert m$^7$G to an abasic site. The resulting abasic site is detected by a specific signature misincorporation during reverse transcription. We show that measurement of N7G accessibility reveals broad insights into RNA tertiary structure. To reveal unexpected conformations states and dynamics of RNA G-quadruplexes in vitro and in cells, we developed DAGGER (Deconvolution Assisted by N7G Gaps Enabled by Reduction and depurination). DAGGER is a computational pipeline that incorporates BASH MaP N7G accessibility data to model RNA structure and identify alternative conformations. Using BASH MaP and DAGGER, we demonstrate that the non-fluorescent conformation of the fluorogenic aptamer Spinach is marked by an alternative tertiary structure that disrupts the folding of Spinach's G-quadruplex. We show that the tertiary structure information afforded by BASH MaP allows substantially improved RNA structure prediction, especially for RNAs with G-quadruplexes. Overall, BASH MaP combines structural information from both the Watson-Crick and Hoogsteen face to reveal RNA tertiary structures and conformational dynamics.

## Results

### BASH MaP reveals DMS-induced m$^7$G sites in RNA

DMS MaP is a popular method for determining RNA structure that primarily reveals the location of RNA helices. However, many tertiary RNA structures cannot be determined using DMS MaP because they often rely on hydrogen bonding with the N7 position of G (N7G; *Vicens and Kieft, 2022*). In principle, the accessibility of N7G can be readily assessed because the N7 position of G is the most nucleophilic site in RNA and therefore the most reactive atom in RNA for DMS methylation (*Lawley and Brookes, 1963*). However, the methyl moiety in m$^7$G is not on the Watson-Crick face and therefore does not readily induce misincorporations during reverse transcription (*Mitchell et al., 2023*).

We therefore sought to make m$^7$G detectable during reverse transcription. Previous studies showed that m$^7$G can be selectively converted to an abasic site, which is an efficient inducer of misincorporations (*Küpfer et al., 2007*). In this reaction, m$^7$G is reduced with potassium borohydride (*Zhang et al.,*

*2022*) and then converted to an abasic site by heating in an acidic buffer (*Enroth et al., 2019*; *Zhang et al., 2019a*; *Figure 1a*). Thus, m$^7$G sites in DMS-treated RNA can be converted to abasic sites, which readily induces misincorporations during reverse transcription (*Zhang et al., 2022*; *Enroth et al., 2019*). Here we refer to BASH MaP as the treatment of RNA with DMS, conversion of m$^7$G to abasic sites, and finally reverse transcription to generate misincorporations at m$^1$A, m$^3$C, and m$^7$G (*Figure 1b*).

We first characterized the misincorporations induced by m$^7$G-derived abasic sites. To test this, we used the 18 S ribosomal RNA (rRNA) from HeLa cells, which contains a stoichiometric m$^7$G at position 1638 (*Piekna-Przybylska et al., 2008*; *Choi and Busch, 1978*; *Haag et al., 2015*). We incubated total HeLa RNA, which contains 18 S rRNA, with potassium borohydride (800 mM) for 30 min to 4 hr and then subsequently heated the RNA at 45 °C in a pH 2.9 buffer for 4 hr to induce depurination. The BASH-MaP-treated HeLa RNA was then reverse transcribed by two commonly used reverse transcriptases in DMS MaP, SuperScript II and Marathon RT, and the 18 S RNA was sequenced.

A plot of misincorporation rate versus reduction duration of DMS treatment showed a~70% misincorporation rate at G1638 after 4 hr of reduction and depurination (*Figure 1c*). This high misincorporation rate is consistent with essentially complete depurination of the m$^7$G based on misincorporation rates seen with synthetic RNA oligos containing abasic sites (*Zhang et al., 2022*). The misincorporations were primarily G→T and G deletions when using SuperScript II (*Figure 1d*). Similar results were seen with Marathon RT, with fewer deletions (*Figure 1—figure supplement 1a*). Together, these data demonstrate that m$^7$G can be efficiently converted to an abasic site and then detected by a misincorporation signature using standard reverse transcription conditions used for DMS MaP.

We next asked whether BASH MaP detects m$^7$G generated by DMS treatment of RNA. For these experiments, we chose the fluorogenic RNA aptamer Spinach (*Paige et al., 2011*). The crystal structure of Spinach revealed a non-canonical G-quadruplex, G's in helices, as well as G's not involved in structural interactions (*Huang et al., 2014*; *Warner et al., 2014*; *Fernandez-Millan et al., 2017*). The N7G of each G in the G-quadruplexes participates in Hoogsteen-face hydrogen-bonding interactions with an adjacent G and is thus expected to be inaccessible to DMS.

We first modified Spinach with DMS (170 mM) for 8 min at 25 °C. DMS-modified Spinach was then split into two fractions and either subjected to BASH MaP or directly reverse transcribed as in DMS MaP. In BASH MaP, misincorporations were detected at G's, which were predominantly G→T and G deletions (*Figure 1d*). The misincorporation signature for these G's matches the misincorporations observed at the m$^7$G site in the 18 S rRNA, thus demonstrating that DMS generated m$^7$G in Spinach.

In contrast, in DMS MaP, misincorporations were rare at G's, and the few misincorporations were predominantly G→A (*Figure 1d*). These misincorporations may reflect mutations introduced during RNA synthesis by T7 RNA polymerase, which are often G→A mutations (*Cheng et al., 2017*). Additionally, the misincorporations may be due to an m$^7$G tautomeric state which induces G→A misincorporations at very low rates during reverse transcription (*Mitchell et al., 2023*; *Štoček and Dračínský, 2020*). Notably, BASH-MaP-treated Spinach produced an average G misincorporation rate of 4.6% compared to 0.58% for DMS MaP and 0.31% for untreated RNA (*Figure 1e*). Altogether, these results demonstrate that BASH MaP detects DMS-mediated methylation of N7G.

DMS is typically used at a final concentration between 10 mM and 170 mM, depending on the level of methylation that is desired (*Choi and Busch, 1978*; *Rouskin et al., 2014*). We therefore modified Spinach with various concentrations of DMS and performed BASH MaP. In addition to m$^1$A, m$^3$C, and m$^7$G, DMS also forms m$^1$G and m$^3$U to a lesser extent (*Mustoe et al., 2019*; *Mitchell et al., 2023*). The misincorporation rates for all four nucleotides were highly correlated between 85 mM and 170 mM DMS ($R^2=0.9928$) with a high Pearson correlation coefficient over a range of DMS concentrations (*Figure 1f* and *Figure 1—figure supplement 1b*). Thus, BASH MaP produces highly reproducible measures of DMS reactivity over a range of DMS concentrations.

We next asked whether the reduction and depurination steps in BASH MaP impaired detection of other methylated nucleotides that are formed in single-stranded RNA. Of the five DMS-modified nucleotides, only m$^7$G fails to efficiently induce misincorporations following reverse transcription (*Vicens and Kieft, 2022*). To test whether BASH MaP conditions affect the detection of m$^1$A, m$^3$C, or m$^3$U, we modified Spinach with DMS (170 mM) for 8 min at 25 °C in bicine buffer at pH 8.3, which increases the formation of m$^3$U (*Mustoe et al., 2019*; *Olson et al., 2022*). DMS-modified Spinach was then split into two fractions and either subjected to BASH MaP or directly reverse transcribed as

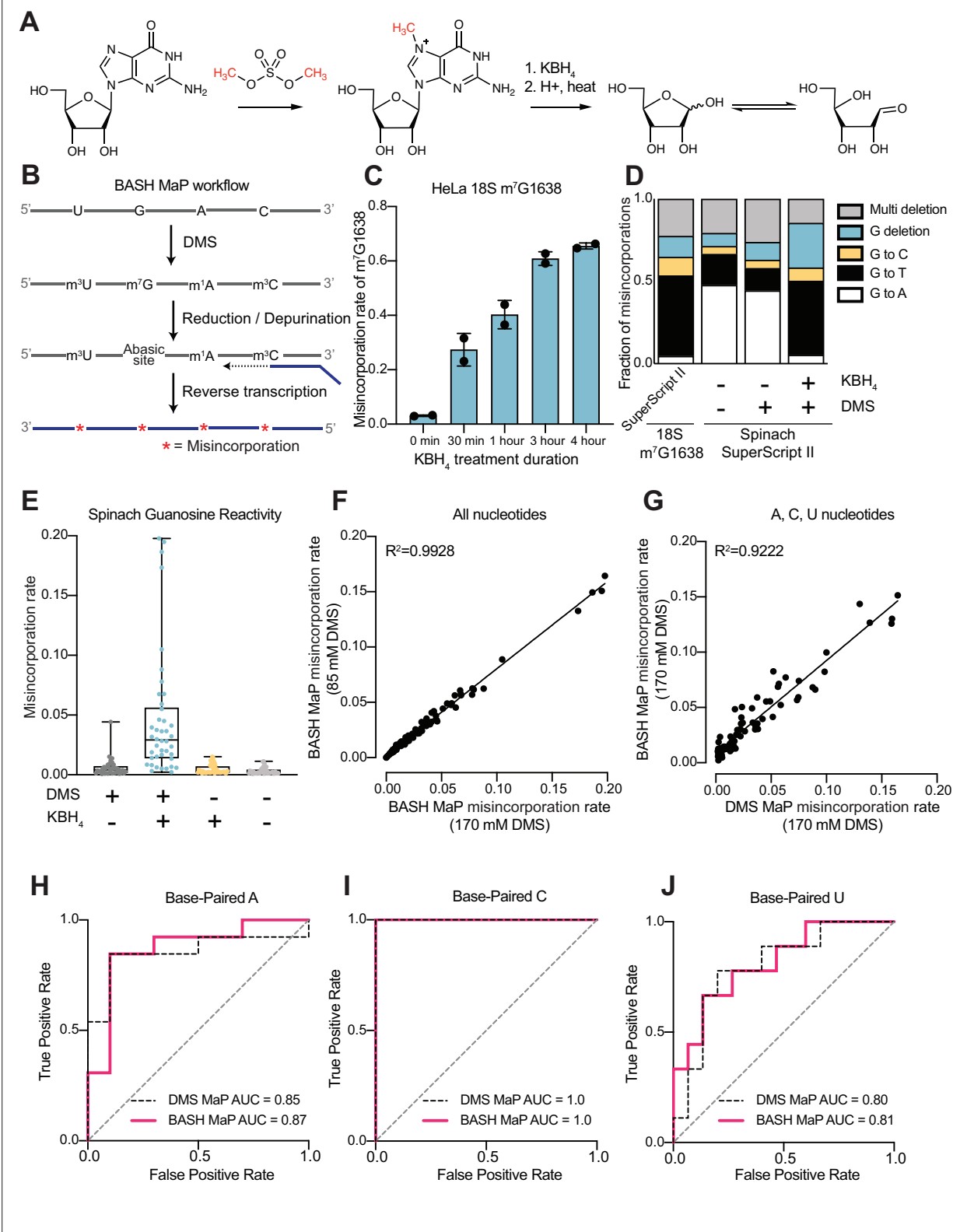

**Figure 1.** Reduction and depuration of DMS-modified RNA enables detection of N7G-DMS adducts with misincorporations. (**A**) Strategy for the misincorporation detection of N7G methylation. N7-methylated G is reduced by potassium borohydride and then heated in acidic conditions to yield an abasic site. The abasic site then proceeds to induce misincorporations in the cDNA following reverse transcription. (**B**) Overall schematic for the BASH MaP experimental workflow. RNA is first treated with dimethyl sulfate (DMS) which produces the following adducts: m1A, m3C, m7G, and to

*Figure 1 continued on next page*

*Figure 1 continued*

a lesser extent m$^1$G and m$^3$U. DMS-modified RNA is then subjected to reduction by potassium borohydride (800 mM) for 4 hr at room temperature followed by purification and heating in a pH 2.9 buffer of acetic acid and sodium acetate for 4 hr at 45 °C. RNA is then purified and subjected to reverse transcription with enzymes and buffer conditions which promote cDNA misincorporations at methylated bases. (**C**) Optimization of reduction duration and efficiency of abasic sites to induce misincorporations. To determine the optimal reduction duration, we treated total HeLa RNA with potassium borohydride (800 mM) for various durations of time, performed depurination for 4 hr in acid buffer, and then prepared RT-PCR amplicons surrounding the endogenously methylated base m$^7$G1638 in the 18 S rRNA. We then quantified the fraction of reads containing a misincorporations at G1638 divided by the total number of reads to yield a misincorporations rate at G1638 following amplicon sequencing. Plots of misincorporations rates revealed that 4 hr of borohydride treatment induced an average misincorporations rate of 70% at G1638. (**D**) Misincorporation signature of abasic sites under reverse transcription conditions for detection of methylated bases. On the left, quantification of the types of misincorporations at G1638 as described in (**C**) for SuperScript II, a reverse transcriptase enzyme commonly used to detect methylated bases with cDNA misincorporations. On the right, fraction of each type of misincorporation calculated collectively from all G residues in Spinach following reverse transcription with SuperScript II. Spinach RNA was either modified with DMS (170 mM) for 8 min at 25 °C or treated with an ethanol control. Modified and control RNA was then reduced with potassium borohydride (800 mM) for 4 hr or incubated in water for 4 hr. All three Spinach samples underwent identical heating in acidic buffer conditions before undergoing reverse transcription. Comparison of types of misincorporations shows that reduction of DMS-treated Spinach RNA produces a misincorporation signature at G residues which mirrors the positive control G1638 when reverse transcribed with SuperScript II. (**E**) Reduction of DMS-treated Spinach RNA produces novel misincorporation data at G bases. To determine if Spinach is highly modified by DMS at N7G, we utilized the experimental data as described in (**E**) with an additional control group in which DMS was omitted but the sample underwent reduction and depurination. Not shown, all four samples underwent identical heating in acidic buffer prior to reverse transcription. We then plotted the misincorporation rate of each G in Spinach for each experimental condition. This misincorporation rate reveals a dramatic increase in misincorporation rates for G bases in Spinach modified with DMS and reduced with potassium borohydride. (**F**) Reproducibility of BASH MaP. Spinach RNA was probed with either 85 mM or 170 mM DMS for 8 min at 25 °C and then reduced and depurinated. The misincorporation rate at each position in Spinach was compared between the two samples and a linear regression was performed which showed an R$^2$ of 0.9928 demonstrating high reproducibility. (**G**) Effect of reduction and depurination on the detection of m$^1$A, m$^3$C, and m$^3$U. To determine whether reduction and depurination of DMS-treated RNA impaired the detection of other methylated bases, we treated Spinach with DMS (170 mM) for 8 min at 25 °C using buffer conditions which promote the methylation of m$^1$G and m$^3$U (***Mustoe et al., 2019***; ***Mitchell et al., 2023***). Then, DMS-treated Spinach was either directly reverse transcribed (DMS MaP) or subjected to reduction and depurination (BASH MaP) before reverse transcription. We then compared the misincorporation rate at each A, C, and U position in Spinach and performed a linear regression. The R$^2$ of 0.9222 demonstrates that reduction and depurination do not impair the detection of m$^1$A, m$^3$C, and m$^3$U generated by the modification of RNA by DMS. (**H–J**) Receiver operator characteristic curves demonstrate that BASH MaP identifies single-stranded regions of RNA. To determine if BASH MaP could accurately distinguish single-stranded from base-paired A, C, and U bases, we constructed Receiver Operator Characteristic (ROC) curves for A, C, and U bases for both BASH MaP and DMS MaP as described in (**G**). The larger the AUC, the better a method is at discriminating paired vs unpaired RNA bases. An AUC = 1.0 demonstrates perfect discrimination ability. Panels (**H–J**) demonstrate that BASH MaP accurately discriminates between single-stranded and base-paired A, C, and U bases.

The online version of this article includes the following figure supplement(s) for figure 1:

**Figure supplement 1.** Comparison of reverse transcriptases on the misincorporation signature of abasic sites and BASH MaP reproducibility and correlation with DMS MaP.

in DMS MaP. Notably, the misincorporation rate of A, C, and U bases remained correlated between the two samples (***Figure 1g*** and ***Figure 1—figure supplement 1b***). We could not measure m$^1$G in BASH MaP as it is obscured by the high rate of m$^7$G formation (***Figure 1—figure supplement 1b***). As expected, A, C, and U bases with high misincorporation rates predominantly mapped to single stranded regions (***Figure 1h–j***). Together, these results demonstrate that the reduction and depurination steps of BASH MaP do not compromise the detection of methylated A, C, and U.

## BASH MaP identifies guanosines that form G-quadruplexes

We next wanted to determine the misincorporation rate of G's located in different structural contexts. Spinach contains G's involved in Hoogsteen interactions involving the N7G, as well as G•C, G•U, and G•A base pairs. Spinach also displays a complex tertiary fold involving a G containing a hydrogen-bonded N7 in a mixed tetrad beneath the two G-tetrads (***Huang et al., 2014Warner et al., 2014***; ***Figure 1—figure supplement 1c***).

We performed BASH MaP on Spinach (1 µM) in solution with its cognate ligand DFHBI-1T (5 µM). Although m$^7$G in BASH MaP can exhibit either misincorporations or deletions, we only quantified misincorporations. We chose to ignore all deletions because a deletion within a stretch of two or more consecutive G's cannot be assigned to any specific G. We converted misincorporation rates for each G into DMS reactivity values through a previously described normalization scheme (***Mitchell et al., 2023***). This approach allows DMS reactivities to be binned as either low, medium, or high. The calculation of DMS reactivity values also enables direct comparison between different nucleotides

which display differences in baseline reactivity to DMS (*Mitchell et al., 2023*). We overlayed the DMS reactivity values on a secondary structure model of Spinach which revealed a complex pattern of N7G reactivity to DMS (*Figure 2a*).

We first examined G's that are not base-paired and are not hydrogen bonded at the N7 position in the Spinach crystal structure. A total of three such G's are found in Spinach and an adjacent stem loop, which is introduced for DNA sequencing (*Merino et al., 2005*; *Figure 2—figure supplement 1a*). For the following comparisons, we directly used misincorporation rates instead of normalized DMS reactivities values. Each of these three G's displayed a high misincorporation rate (*Figure 2b*). Thus, as expected, unpaired G's with free N7 positions display high N7G reactivity to DMS.

In contrast, most G's in helices exhibited reduced misincorporation rates compared to G's in single-stranded regions (*Figure 2b*). Although helical G's do not have hydrogen-bonds at the N7 position, the reduced methylation of helical G's is consistent with low accessibility of the major groove in duplex RNA (*Weeks and Crothers, 1993*). However, we noticed two base-paired G's, G81 and G102, with very high misincorporation rates (*Figure 2a–b*). These G's are located at helix termini, a location previously shown to exhibit enhanced major groove accessibility, and in locations of unusual base stacking (*Weeks and Crothers, 1993*). These results suggest that base-paired G's show moderately low N7 reactivity to DMS except when located at the end of a helix.

We next asked whether N7-hydrogen bonds block methylation by DMS. We annotated all G's with N7 positions engaged in a hydrogen bond, which included G's in G-quadruplexes and the G involved in the mixed tetrad (*Huang et al., 2014*; *Warner et al., 2014*). We found that these G's showed the lowest misincorporation rates compared to base-paired G's and single-stranded G's (*Figure 2b*). Notably, the G's constituting the RNA G-quadruplex of Spinach showed the lowest misincorporation rates in Spinach (*Figure 2c*). Thus, G's that are highly resistant to DMS methylation are likely engaged in hydrogen-bonding interactions involving N7 and may reflect G's within G-quadruplexes.

To further test whether BASH MaP can identify G's in G-quadruplexes, we performed BASH MaP on a polyUG repeat RNA which was recently shown to adopt an atypical G-quadruplex (*Roschdi et al., 2022*). We choose a UG repeat length of 15 because we reasoned that this repeat length could theoretically form four unique G-quadruplexes composed of 12 consecutive GU repeats (*Figure 2c*). The formation of four unique G-quadruplexes would occur because of 'register shifts', a form of alternative RNA tertiary conformations (*Figure 2c*). We reasoned that each unique G-quadruplex register could be differentiated by the reactivity of three G's not engaged in a G-quadruplex (*Figure 2c*). Because these three excluded G's should lack N7 hydrogen bonding, they therefore are expected to display higher DMS-mediated methylation than the G's engaged in a G-quadruplex in all four registers.

BASH MaP analysis of the 15 x polyUG repeat RNA revealed low misincorporation rates for the nine G's expected to be hydrogen bonded in all four G-quadruplex registers (G26-G42) (*Figure 2e*). G's that could participate in only one G-quadruplex showed the highest misincorporation rate (*Figure 2f*). Interestingly, a plot of the average misincorporation rate for each register revealed that the propensity to form each register was not equal (*Figure 2—figure supplement 1b*). Instead, the G-quadruplex in register 2, which starts at G22 and ends at G44, displayed the lowest average misincorporation rate and therefore marks the most abundant register (*Figure 2—figure supplement 1b*). Overall, these experiments demonstrate that BASH MaP enables fine resolution of different types of G-quadruplex structures.

Although we focused on the N7 position of G to identify the location of G-quadruplexes, we also considered the possibility that other positions within G could be useful for G-quadruplex prediction, particularly the N1 position (N1G). N1G is hydrogen bonded in G-quadruplexes and in standard Watson-Crick base pairs and is therefore expected to be protected from DMS methylation. Therefore, N1G reactivity might reveal the location of G-quadruplexes.

Compared to the N7G, we found the reactivity of N1G is between 10- and 100-fold lower, even under higher pH conditions which promote the formation of $m^1G$ (*Mustoe et al., 2019*; *Mitchell et al., 2023*; *Figure 2—figure supplement 1c*). Nevertheless, with sufficient sequencing depth, the presence of $m^1G$ can be detected by signature G→C and G→U misincorporations (*Mitchell et al., 2023*). Using this approach, we observed no discrimination between base-paired G's and G's involved in a G-quadruplex (*Figure 2—figure supplement 1c*). Thus, the reactivity values of N1G are insufficient to identify G's in a G-quadruplex. Overall, these data suggest that N7G reactivity measured by BASH MaP uniquely identifies G's that are involved in a G-quadruplex.

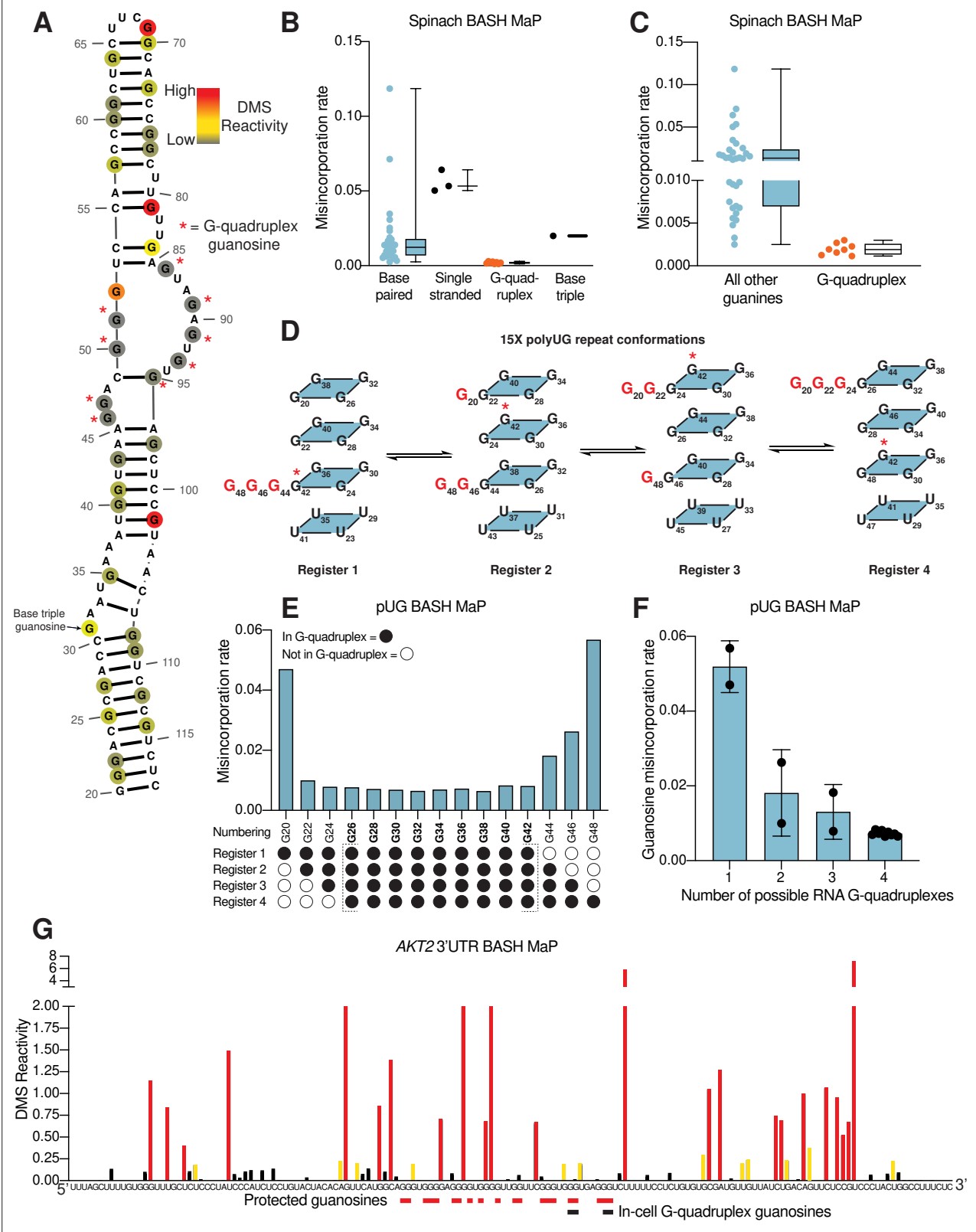

**Figure 2.** N7G reactivity reveals guanosines involved in tertiary interactions. (**A**) N7G reactivity overlaid on the secondary structure model of Spinach as derived from the crystal structure 4TS2 (***Warner et al., 2014***). G-quadruplex G's are indicated with a red asterisk. N7G reactivity is colored on a continuous gradient from grey (low reactivity) to yellow (moderate reactivity) to red (high reactivity). N7G reactivities represent normalized values derived from misincorporation rates. Spinach BASH MaP data was obtained as described in ***Figure 1G*** and no background subtraction was utilized.

*Figure 2 continued on next page*

*Figure 2 continued*

(**B**) Comparison between structural features in Spinach and observed BASH MaP misincorporation rate at G bases. To determine which structural features impact N7G methylation rate we utilized the crystal structure 4TS2, and classified individual G's as either base-paired, single-stranded, engaged in an RNA G-quadruplex, or engaging in a base triple. This plot reveals single-stranded G's have relatively high methylation rates whereas RNA G-quadruplex G's are strongly protected from N7G methylation. (**C**) Comparison between RNA G-quadruplex G's and all other G's in Spinach reveals RNA G-quadruplex G's are the most protected G's from N7G methylation in Spinach. (**D**) Schematic of alternative conformations the 15 x polyUG (pUG) repeat RNA can adopt in solution. The 15 x pUG repeat can adopt four distinct RNA G-quadruplex conformations of 12 consecutive GU repeats. Each of these four distinct conformations is indicated as Register 1 through 4. Each register is uniquely defined by a set of three G's not engaged in an RNA G-quadruplex (colored red). These G's are predicted to be single stranded or base-paired and thus display much higher reactivity than the G's engaged in the RNA G-quadruplex. To show how the G-quadruplex is rearranged in each conformation, G42 is labeled with a red asterisk. (**E**) BASH MaP misincorporations rate plot of pUG G's. 15 x pUG RNA was refolded in potassium buffer (100 mM) and modified with DMS (170 mM) for 6 min at 37 °C. G's engaged in a G-quadruplex for each unique register are indicated at the bottom of the plot. G's predicted to be engaged in a G-quadruplex in each of the four registers is bolded and bracketed below the plot. The misincorporation rate plot reveals that G's engaged in a G-quadruplex in all four registers display a strong protection from DMS methylation. Consequently, G20 and G48, which are only predicted to be protected in a single register show the highest misincorporation rates. (**F**) Quantification of (**E**) shows the relationship between G inclusion in RNA G-quadruplex registers and measured misincorporation rate. The plot reveals increased protection from N7G methylation for G's as the number of G-quadruplex conformations increases. (**G**) BASH MaP of a G-quadruplex in the 3'UTR of the *AKT2* mRNA. SH-SY5Y cells were treated with DMS (170 mM) for 6 min at 37 °C. Primers were designed to specifically amplify the putative G-quadruplex region in the *AKT2* mRNA. Misincorporation rates were converted to normalized reactivity values as described in METHODS. Putative G-quadruplex G's as identified in rG4-seq (*Warner et al., 2014*) are highlighted in red. Red bars below the sequence indicate putative G-quadruplex G's with strong protections from N7G-DMS methylations. G's previously identified as engaged in a G-quadruplex in cells are indicated below with a black bar. The normalized reactivity plot reveals N7G protections from DMS at previously identified in cell G-quadruplex G's as well as other G-tracts. Together, these data support the formation of a G-quadruplex in 3'UTR of the *AKT2* mRNA in SH-SY5Y cells.

The online version of this article includes the following figure supplement(s) for figure 2:

**Figure supplement 1.** DMS MaP is unable to efficiently detect accessibility of the N7 position in guanine and validation of *AKT2* N7G protections in a detected G-quadruplex.

## BASH MaP detects G-quadruplexes in cellular mRNAs

The above experiments examined RNA G-quadruplexes in vitro. We next asked whether BASH MaP could identify G's involved in an RNA G-quadruplex in cells. Methods for mapping G-quadruplexes in cells are limited. One method involves treating cells with DMS and then utilizing the property of folded G-quadruplexes to induce reverse transcriptase stops (RT stops) at their 3' ends only in potassium-rich buffers (*Guo and Bartel, 2016*; *Kwok et al., 2016a*). Cells are first treated with DMS, which modifies any N7G's that are not in a G-quadruplex, and then the RNA is harvested and refolded in potassium buffer, and in parallel, a buffer containing either sodium or lithium. If the G-quadruplex was folded in the cell then its N7G's should be protected from methylation and the G-quadruplex would readily reform when folded in potassium buffer. If the G-quadruplex was unfolded, then the N7G's would become methylated which prevents subsequent G-quadruplex refolding. The refolded G-quadruplexes are then identified as potassium-dependent RT stops and represent G-quadruplexes in cells.

A second method is based on the propensity for terminal G-quadruplex G's to become chemically acylated in cells by a class of RNA-modifying chemicals called SHAPE reagents (*Kwok et al., 2016b*). The location of SHAPE reagent adducts in extracted cellular RNA is tested by reverse-transcription stops in potassium free buffers. Together, these methods predict the 3' end of G-quadruplexes in cells. However, these G-quadruplex mapping methods only infer the presence of a G-quadruplex and cannot reveal multiple G-tetrads or the presence of atypical G-quadruplexes (*Kwok et al., 2016b*).

Despite their low resolution, these methods have suggested that most G-quadruplexes in mRNAs are globally unfolded in HEK293T and HeLa mammalian cell lines (*Guo and Bartel, 2016*; *Kharel et al., 2023*). Although most G-quadruplexes are unfolded, a small subset of G-quadruplexes appeared to remain folded in cells. One of these folded G-quadruplexes resides in the 3' UTR of the *AKT2* mRNA (*Guo and Bartel, 2016*; *Figure 2—figure supplement 1f*). We therefore sought to use BASH MaP to validate the folding status of the *AKT2* 3' UTR G-quadruplex.

To determine if the 3' UTR of the *AKT2* mRNA contains a G-quadruplex in cells, we treated SH-SY5Y cells with DMS (170 mM) for 6 min at 37 °C (*Mitchell et al., 2023*; *Olson et al., 2022*). We then performed BASH MaP on the extracted total RNA. The putative folded G-quadruplex region of *AKT2* was PCR amplified and sequenced to identify misincorporation rates at G's. Notably, the

misincorporations at G's in *AKT2* reflect formation of m$^7$G since they exhibited the characteristic G→T and G deletion misincorporation signature of m$^7$G in BASH MaP (*Figure 2—figure supplement 1e*).

Next, we asked whether G's which were previously identified as the 3' end of the *AKT2* folded G-quadruplex displayed low DMS reactivity. We converted misincorporation rates into normalized DMS reactivities and plotted the reactivity profile (*Mitchell et al., 2023*; *Figure 2g*). The in-cell G-quadruplex folding data suggests that the *AKT2* 3' UTR G-quadruplex has multiple conformations with four unique 3' ends identified by potassium-dependent RT stops (*Figure 2g* and *Figure 2—figure supplement 1e*). Within the putative G-quadruplex region, we were able to distinguish several tracts of G's with very low DMS reactivity, reflecting N7 positions protected from DMS (*Figure 2g*). These protected G's mapped specifically to G's previously identified to be the 3' end of the in-cell folded G-quadruplex (*Figure 2g* and *Figure 2—figure supplement 1e*). Together, these results suggest that BASH MaP can identify G's engaged in a G-quadruplex in cells.

In principle, a G-quadruplex only comprises four tracts of G's. However, *AKT2* contains seven tracts of three or more G's which show low reactivity. Therefore, the BASH MaP N7G accessibility data is unclear about which tracts of G's are engaged in the G-quadruplex (*Figure 2g*). However, as will be described below, specific G-quadruplex G's can be identified using a method that involves assessing co-occurring methylation events. The *AKT2* mapped G-quadruplex region contains several tracts of G's with high reactivity to DMS, which reflects accessible N7Gs and suggests that these G tracts are not engaged in a G-quadruplex (*Figure 2g* and *Figure 2—figure supplement 1e*). The location of these highly reactive G tracts suggests that the *AKT2* G-quadruplex may adopt an unusual topology.

## The G-quadruplex core of Spinach is marked by co-occurring G – G misincorporations

In addition to providing information on nucleobase accessibility, DMS MaP can predict nucleobase-nucleobase interactions due to the phenomenon of 'RNA breathing' (*Homan et al., 2014*). During RNA breathing, a transient local unfolding event can lead to methylation of an otherwise inaccessible nucleotide. The methylated nucleotide can no longer interact with its cognate nucleotide partner, which leads to methylation of the nucleotide partner as well. These events produce pairs of misincorporations that repeatedly co-occur on the same strand of RNA (*Mustoe et al., 2019*; *Homan et al., 2014*; *Olson et al., 2022*; *Krokhotin et al., 2017*). By using a statistical analysis of misincorporations co-occurring in individual sequencing reads, with each read representing a unique RNA molecule (*Homan et al., 2014*), Watson-Crick base pairs have been identified in cells (*Mustoe et al., 2019*). We therefore wondered whether BASH MaP can identify the specific G's in G-quadruplexes using a similar statistical analysis of co-occurring misincorporations between G's.

We first asked whether BASH MaP produced enough co-occurring misincorporations needed for statistical analysis. Statistical analysis of co-occurring misincorporations requires two or more misincorporations in a sequencing read, which in DMS MaP is limited to misincorporations induced by m$^1$A and m$^3$C due to low rates of m$^1$G and m$^3$U formation (*Homan et al., 2014*). Analysis of Spinach BASH MaP data indicated significantly more misincorporations per RNA molecule than DMS MaP (*Figure 3—figure supplement 1a*). The percentage of reads with two or more misincorporations doubled after BASH MaP, and the average number of total misincorporations per read increased from 1.35 to 2.9 (*Figure 3—figure supplement 1b–c*). These results suggest that BASH MaP produces data which is ideal for the statistical analysis of co-occurring misincorporations.

Statistical analysis of co-occurring misincorporations is performed with the program RING Mapper which identifies pairs of nucleotides which display higher than expected rates of co-misincorporations (*Mustoe et al., 2019*; *Homan et al., 2014*). RING Mapper utilizes a likelihood-ratio statistical test called a G-test to calculate whether a chemical modification event at one location affects the probability of a chemical modification at a second location. If a chemical modification at a position in an RNA increases the probability of a chemical modification at a second location, then the two locations display statistical dependence. In contrast, if a chemical modification at one position has little or no effect on the probability of modification at a second location, then the two locations display statistical independence. Nucleotide pairs with a high G-test statistic value indicate high dependence and are thought to represent pairs of nucleotides which interact in the RNA structure (*Mustoe et al., 2019*; *Homan et al., 2014*).

We next asked whether G's in BASH-MaP-treated Spinach RNA produced patterns of co-occurring misincorporations that cannot be seen in DMS MaP. To identify novel patterns of co-occurring misincorporations in BASH MaP of Spinach, we calculated G-test statistic values with RING Mapper and converted these to a normalized correlation strength value. We then compared these values to annotated domains derived from the crystal structure of Spinach (; *Figure 3c*). A heatmap of normalized correlation strength for all possible combinations of base pairs showed marked differences between BASH-MaP-treated Spinach and control DMS-MaP-treated Spinach (*Figure 3d–e*).

We next asked whether the patterns of co-occurring misincorporations produced by BASH MaP represent structural interactions in Spinach which are invisible to DMS MaP. DMS-MaP-treated Spinach displayed strong correlations between A's in the P2 domain and G-quadruplex G residues but displayed no correlations between G-quadruplex G's (*Figure 3d*). In contrast, BASH-MaP treated Spinach produced G – G correlations between G's comprising the G-quadruplex (*Figure 3e*). These G – G correlations represent the hydrogen bonding of N7Gs engaged in a G-quadruplex. We previously relied on measures of N7G reactivity to infer G-quadruplex G's (*Figure 2*). Instead, the co-occurring misincorporations heatmap suggests that the G – G correlations in BASH MaP can directly identify the specific G's engaged in a G-quadruplex in an unbiased manner. Thus, G-quadruplex G's can be discovered using BASH MaP, in contrast to DMS MaP and produce a specific signature in the RING MaP heatmap.

Since BASH MaP produced many G – G co-occurring misincorporations, we next wondered whether network analysis could identify discrete groups of structurally linked G's in Spinach such as G-quadruplex G's. We reasoned that since the G's in the Spinach G-quadruplex form a hydrogen-bonding network, then the G-quadruplex G's would form a distinct cluster in a network analysis of G – G co-occurring misincorporations (*Figure 3c*). To visualize the relationships between G – G co-occurring misincorporations, we first represented the collection of co-occurring misincorporations between pairs of G's as a network where vertices represent unique G's in Spinach and edges denote co-occurring misincorporations (*Sengupta et al., 2019*.). To remove low confidence co-modified nucleotides, we performed Z-score normalization on the collection of G-test statistic values. We then filtered the network by Z-scores and included only co-occurring misincorporations with a Z-score above a certain threshold.

To identify discrete groups of structurally linked G's in Spinach, we performed a detailed visual analysis of the BASH MaP G – G co-occurring misincorporations network. The BASH MaP network revealed a substantially higher number of co-occurring misincorporations between G nucleotides compared to DMS MaP (*Figure 3f–g*). We implemented a Z-score cutoff of 2.0 which resulted in the formation of two distinct clusters within the network (*Figure 3g*). The smaller cluster was composed of seven G nucleotides, with six of them forming part of the Spinach G-quadruplex and mixed tetrad (*Figure 3g*). Notably, three G-quadruplex G's: G50, G51, and G89, were not represented in the network because they did not display co-occurring misincorporations with other G's. The lack of co-occurring misincorporations suggests that methylation of G50, G51, and G89 produces an RNA conformation which retains hydrogen bonds at the N7G of other G-quadruplex G's. Instead, G97, which is base stacked below G95 of the mixed tetrad, was present in the second, smaller cluster (*Figure 3g*). These results show that network analysis of BASH MaP data enables visualization of structurally linked groups of G's and revealed that G-quadruplex G's form a distinct cluster.

Network analysis of G – G co-occurring misincorporations also revealed a large cluster of G's which are annotated as base paired (*Figure 3g*). The network is evident as a diffuse checkerboard pattern in the RING MaP heatmap for G nucleotides engaged in helical regions outside of the G-quadruplex core (*Figure 3—figure supplement 1d*). These results reveal that G's engaged in canonical base pairs contribute to the stability of nearby base-paired G's.

In some cases, base-paired G's displayed moderate levels of co-methylation with other base-paired G's regardless of if the G's were in the same or different helix (*Figure 3—figure supplement 1d*). The mechanism for these co-occurring methylations is unclear but suggests that methylation of even a single N7G within Spinach can lead to global destabilization of RNA structure. Together, these results further show that BASH-MaP-treated RNA produces novel patterns of co-occurring methylations which correspond to structurally linked G's.

We next asked if G – G co-occurring misincorporations and network analysis could more confidently identify G-quadruplex G's in *AKT2*. We performed RING MaP analysis of *AKT2* BASH MaP data and

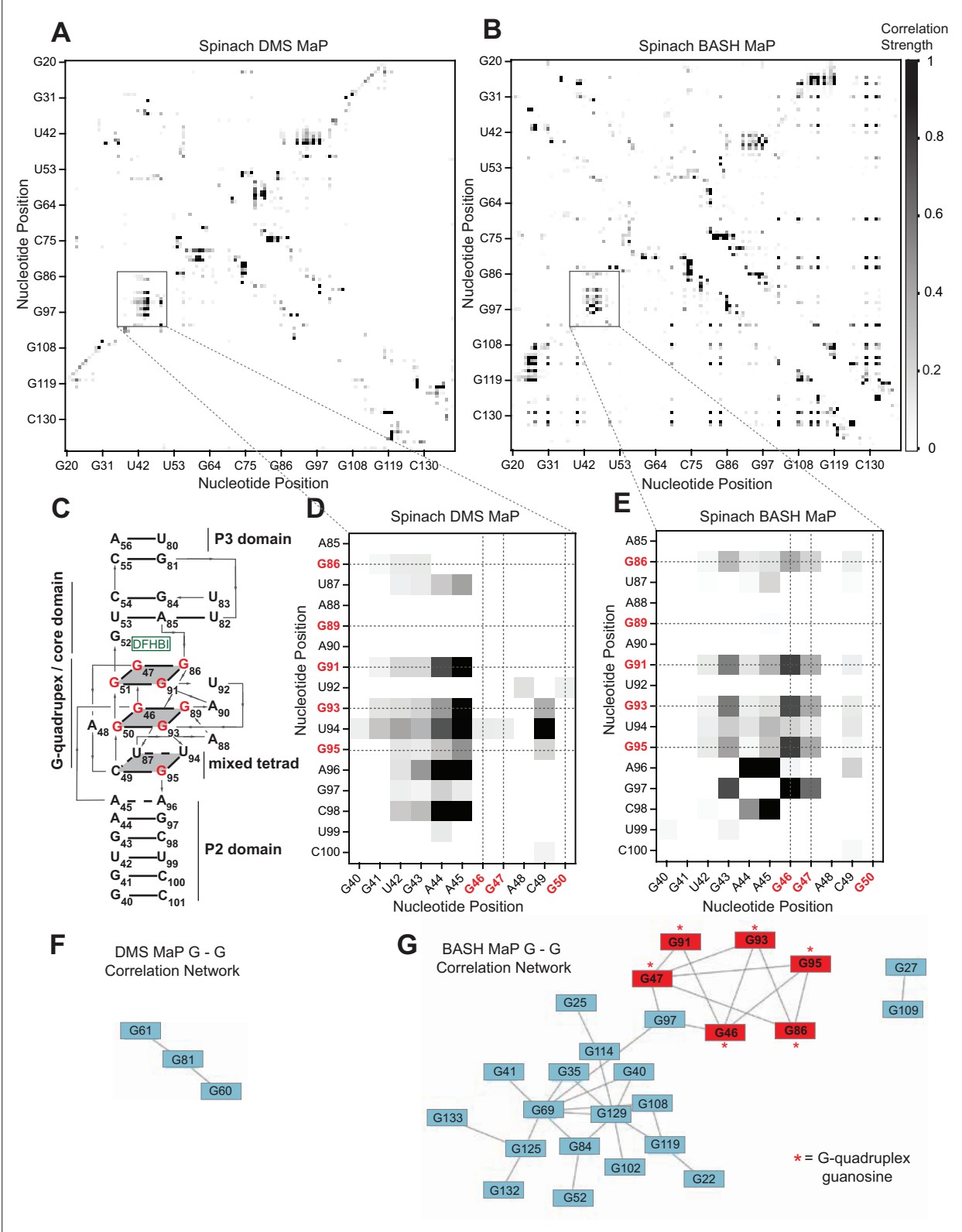

**Figure 3.** BASH MaP reveals networks of co-occurring modifications in the Spinach G-quadruplex. (**A, B**) Heatmap of correlation strength between misincorporation that co-occur on the same sequencing read for Spinach treated with DMS MaP (**A**) or BASH MaP (**B**). Spinach was treated with DMS (170 mM) for 8 min at 25 °C. Each point represents a G-test significance value as calculated by RING Mapper which was then scaled to a value between zero and one. Values closer to one appear darker in the correlation heatmap and represent higher G-test correlation strength. (**C**) Three-dimensional

*Figure 3 continued on next page*

*Figure 3 continued*

model of Spinach core ligand binding domain with numbering scheme used in (**A–D**) and (**F, G**). G-quadruplex and mixed tetrad interactions in Spinach are indicated with a grey plane. The ligand DFHBI-1T interacts with G52 and sits between the upper G-quadruplex tetrad and a U-A-U base triple. Structural domains P2, P3, and the core domain are indicated. (**D**) Close up of boxed region in (**A**) shows the pattern of co-occurring misincorporations surrounding the bases involved in the G-quadruplex of Spinach (marked in red). This heatmap displays predominantly co-occurring misincorporations between A – A, A – C, and A – G positions. (**E**) Close up of boxed region in (**B**) shows the pattern of co-occurring misincorporations surrounding the bases involved in the G-quadruplex of Spinach (marked in red). This heatmap is enriched in co-occurring misincorporations between G – G positions and displays correlations between G's involved in the G-quadruplex of Spinach. (**F, G**) Network analysis of G – G correlations in DMS MaP (**F**) and BASH MaP (**G**) of Spinach. A network was constructed by representing G positions in Spinach as vertices and creating edges between G positions that display co-occurring misincorporations above a Z-score threshold of 2.0. BASH MaP of Spinach reveals that G's form two major clusters. The smaller cluster contains six of nine G's which comprise the Spinach G-quadruplex.

The online version of this article includes the following figure supplement(s) for figure 3:

**Figure supplement 1.** BASH MaP improves single molecule analysis through increased misincorporation density.

**Figure supplement 2.** *AKT2* 3'UTR adopts an atypical G-quadruplex with a long central loop.

represented the G – G co-occurring misincorporations as a network (*Figure 3—figure supplement 2a*). The network formed four main clusters, one of which contained four connected and lowly reactive G's (*Figure 3—figure supplement 2b*). Interestingly, the co-occurring misincorporation network data indicates an unusual G-quadruplex topology where at least one conformation of the *AKT2* G-quadruplex involves the first two and last two G tracts of the putative G-quadruplex region (*Figure 3—figure supplement 2c*). We used the program QGRS mapper (*Kikin et al., 2006*) to predict all possible three-tiered G-quadruplex conformations and identified 129 unique G-quadruplex conformers (*Figure 3—figure supplement 2d*). We then calculated the average misincorporation rate of the G-quadruplex G's for each conformation and ordered the predicted conformers by average misincorporation rate. Analysis of the 10 G-quadruplex conformers with the lowest average misincorporation rate revealed a common large central loop element (*Figure 3—figure supplement 2d*). Overall, these results further support the formation of a G-quadruplex in the *AKT2* 3' UTR in cells and suggests the *AKT2* 3' UTR G-quadruplex may adopt an unusual topology with a long central loop.

## Multiplexed mutagenesis and chemical probing reveal N7G and base stacking interactions

A second way to identify interactions between nucleobases in RNA is to use the mutate-and-map method (*Cheng et al., 2017*; *Cordero et al., 2014*) (M2). In the mutate-and-map method, mutagenic PCR is used to create a pool of RNAs with PCR-derived mutations at every position along the length of the target RNA. DMS MaP is then performed on the pool of PCR-mutated RNAs (M2 DMS MaP), and the data is demultiplexed to identify how a change in nucleotide identity at each position along an RNA affects the global DMS reactivity of that RNA. When a PCR-derived mutation occurs at the position of a base pairing partner to an A or C, the A or C can no longer base pair and becomes accessible to DMS. M2 DMS MaP provides a highly comprehensive screen of how each position in an RNA interacts with other nucleotides in an RNA. However, M2 DMS MaP primarily detects Watson-Crick base pairs in helical regions in an RNA (*Rouskin et al., 2014*). We reasoned that mutate-and-map combined with BASH MaP (M2 BASH MaP) would enable a comprehensive screen of global N7G interactions, thereby revealing G-quadruplex and other tertiary interactions of N7G in RNA.

To identify N7G interactions that contribute to RNA structure, we performed M2 BASH MaP and M2 DMS MaP and compared the resulting heatmaps to identify N7G-specific interactions. We used mutagenic PCR followed by in vitro transcription to randomly introduce point mutations within Spinach and the 15 x polyUG RNA (*Figure 4—figure supplement 1a*). We then adapted the M2 DMS MaP method to include the reduction and depurination steps of BASH MaP (M2 BASH MaP) and created an analysis pipeline to generate the mutate-and-map heatmap.

To create the mutate-and-map heatmap, we only considered sequencing reads which covered the entire length of the RNA. We then considered only a subset of sequencing reads which contained a PCR-derived mutation at one location along the RNA. Then, we calculated the misincorporation rate at all other positions of the RNA from the subset of sequencing reads. We repeated the previously described data analysis process for each position along the RNA. Finally, all the calculated misincorporation rate profiles were stacked vertically to create the mutate-and-map heatmap.

To better visualize changes in nucleotide reactivity due to the presence of a mutation installed during PCR, we analyzed each column of the heatmap separately by Z-score normalization. Each row of the heatmap represents the misincorporation rate profile for RNAs with a PCR-derived mutation at that indicated position. We converted the misincorporation rates in each column to a Z-score. A Z-score normalization strategy helps to reveal at which location a PCR-derived mutation increases chemical reactivity the most for a given base in RNA (*Rouskin et al., 2014*). To restrict our analysis to increases in chemical reactivity, we plotted only positive Z-scores (*Figure 4a–b*). We then compared normalized heatmaps of M2 DMS MaP with M2 BASH MaP.

M2 BASH MaP of Spinach RNA produced a large collection of unique mutate-and-map signals (*Figure 4b*). The most notable of these signals mapped specifically to the N7G-interactions of the G-quadruplex in Spinach. These interactions were absent in the M2 DMS MaP heatmap (*Figure 4a*). Comparison of mutate-and-map heatmaps suggests that M2 BASH MaP can uniquely identify N7G interactions in RNA.

In mutate-and-map heatmaps, mutations installed during PCR typically increase the reactivity of a single position, namely that location's base-paired partner (*Rouskin et al., 2014*). This led us to ask whether individual G-tetrads which stack to form a G-quadruplex could be determined from M2 BASH MaP heatmaps in a manner analogous to base pair identification.

PCR-derived mutations at a G-quadruplex G could potentially increase the reactivity of only G's in the same tetrad (*Figure 4c*). However, a zoomed-in plot of the Spinach M2 BASH MaP heatmap indicated that a mutation installed during PCR in any G-quadruplex G caused increased N7G methylation in all G-quadruplex tetrad layers (*Figure 4d*). A similar pattern was seen with M2 BASH MaP analysis of 15 x polyUG RNA (*Figure 4—figure supplement 1b*). These results suggest that PCR-derived mutations at G bases in the Spinach and polyUG G-quadruplexes cause large scale increases in N7G accessibility of all G-quadruplex G's. Large scale increases in N7G accessibility due to PCR-derived mutations may reflect global destabilization of the G-quadruplex which leads to unfolding. Therefore, M2 BASH MaP revealed that G-quadruplex folding is easily perturbed by changing any G in the G-quadruplex.

In addition to revealing the G-quadruplex-forming residues, the M2 BASH MaP heatmap revealed conformational alterations to helices after point mutations. Helices are typically detected as a diagonal line in the mutate-and-map heatmap, caused by a PCR-induced mutation at one position, and a DMS-induced increase in reactivity at the complimentary nucleotide in the helix (*Figure 4e*). However, helices in M2 BASH MaP appeared as thicker and jagged diagonal lines as compared to the same helix in M2 DMS MaP (*Figure 4f*). The additional signals in the M2 BASH MaP heatmap indicates that PCR-induced mutations in the Spinach helices cause broader changes in DMS reactivity than to simply the complementary nucleotide.

To better understand the origin of these differences, we first asked which pairs of nucleotides displayed novel M2 BASH MaP signals in the P3 stem of Spinach. Enlargement and visual inspection of the P3 stem in the M2 BASH MaP heatmap revealed that most of the novel signals corresponded to increased reactivity to DMS for G residues in the P3 stem (*Figure 4f*).

One explanation for these signals is a transient breathing of the helix leading to increased N7G accessibility as previously mentioned for G – G co-occurring misincorporations in the RING MaP heatmap. However, G – G co-occurring misincorporations in the RING MaP heatmap occurred between G bases in the same and separate helices. In contrast, M2 BASH MaP signals between pairs of G residues were restricted to the same contiguous helix which suggests a different mechanism.

We additionally observed vertical lines in the P3 stem for the M2 BASH MaP heatmap (*Figure 4f*). PCR-derived mutations at different locations in the P3 stem induced increased reactivity to DMS for the same G bases to create these vertical lines. PCR-derived mutations within the P3 stem are expected to create bulges which disrupt helix base stacking (*Davis et al., 2011*). The highly reactive N7G at helix termini, such as at G81, are also located in regions where base stacking interactions are disrupted (*Figure 4g*). The vertical lines in the heatmap suggest that PCR-derived mutations within helices induce local distortions of base-stacking interactions and consequently increased N7G reactivity of nearby G's. Therefore, local distortions of base stacking caused by PCR-derived mutations may explain the novel signals present in the Spinach M2 BASH MaP heatmap. Together, these data further suggest that N7G reactivity is highly influenced by local base stacking interactions.

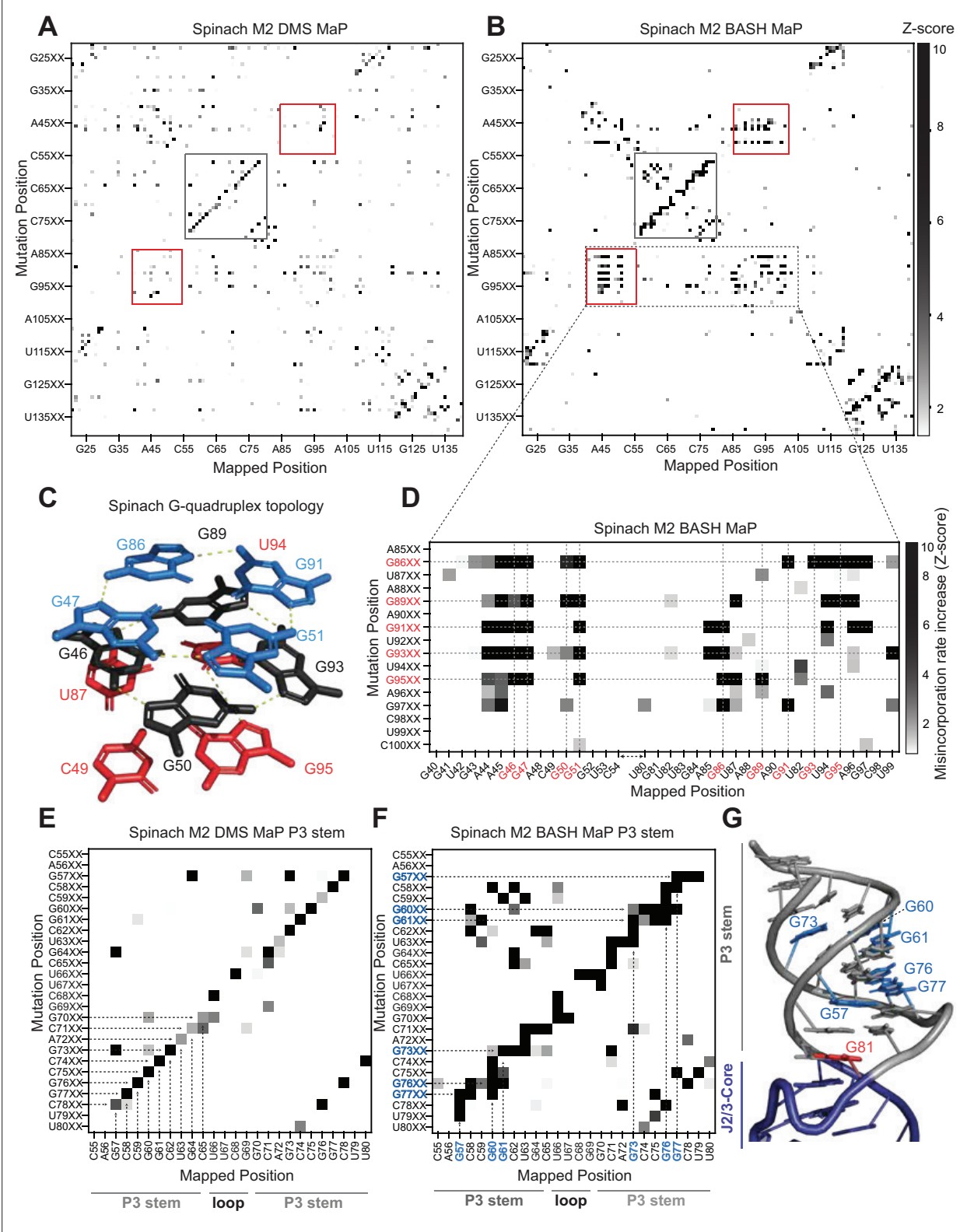

**Figure 4.** Multiplexed probing of single nucleotide mutants identifies N7G and base stacking interactions. (**A–E**) Mutate and Map (M2) heatmaps of DMS MaP (**A**) and BASH MaP (**B**) of randomly mutagenized Spinach RNA. A mutate and map heatmap plots the chemical reactivity profile for RNAs with a PCR-derived mutation at a specific position along the length of the RNA. When a position in an RNA is mutated through mutagenic PCR, it is predicted that all interacting nucleotides will display an increase in chemical reactivity. Each row of the heatmap (Mutation Position) represents

*Figure 4 continued on next page*

*Figure 4 continued*

sequencing reads with a PCR-derived mutation at the indicated position within Spinach. Each column of the heatmap (Mapped Position) represents the misincorporation rate at the indicated position in Spinach. Visualization of changes in reactivity to DMS which are induced by point mutations is enabled by performing Z-score normalizations for each column individually. Only positive Z-scores are plotted to display increases in chemical reactivity due to PCR-derived mutations. M2 BASH MaP displays unique signals in the mutate-and-map heatmap in the G-quadruplex (red box) and P3 (grey box) region of Spinach. (**C**) G-quadruplex topology of Spinach. Spinach contains a two-tiered G-quadruplex with a mixed tetra stacked below. G-quadruplex numberings and colors are used to denote G-tetrads. The top G-tetrad is colored blue. The middle G-tetrad is colored black. The bottom mixed tetrad is colored red. N7G interactions are denoted with a yellow dotted line. (**D**) Mutate-and-Map has previously been used to identify base-pair interactions in RNA. To determine whether N7G interactions could be identified by mutate-and-map we examined the mutate-and-map heatmap generated by M2 BASH MaP of Spinach shown in (**B**) zoomed in on the Spinach G-quadruplex core domain. G-quadruplex G's are indicated in red. The plot reveals that PCR-derived mutations in G-quadruplex G's induce increases in N7G reactivity to DMS throughout the entire Spinach core domain. This suggests that PCR-derived mutations within the Spinach G-quadruplex perturb the structure of the entire core domain. (**E**) Zoom in of the M2 DMS MaP heatmap for the P3 domain of Spinach. P3 stem and loop nucleotides are indicated below the plot. The base-pairing pattern of the P3 domain is clearly identified as a diagonal line in the mutate-and-map heatmap. Base pairs are highlighted with dotted arrows. (**F**) Zoom in of M2 BASH MaP heatmap for the P3 domain of Spinach. G's which display long vertical lines in the heatmap and give the P3 region a jagged appearance are highlighted in blue. The vertical lines indicate that PCR-derived mutations at multiple adjacent nucleotides all cause increased reactivity to DMS of highlighted G's. This suggests that N7G reactivity to DMS is sensitive to local disruption of helix stacking. (**G**) Crystal structure of the Spinach P3 stem. The P3 stem is denoted in grey. G bases which display signals in the M2 BASH MaP heatmap are highlighted in blue. Hyper-reactive G's at helix termini such as G81 are colored red.

The online version of this article includes the following figure supplement(s) for figure 4:

**Figure supplement 1.** M2 BASH MaP of 15 x polyUG repeat RNA.

## Tertiary folding constraints improve RNA secondary structure modeling

Incorporation of chemical probing data greatly improves RNA secondary structure modeling for most RNAs (*Deigan et al., 2009*; *Mathews et al., 2004*; *Cordero et al., 2012*). The secondary structure of RNA refers to the location of base pairs in RNA and is typically used to represent RNA structure (*Vicens and Kieft, 2022*). The locations of base pairs are usually predicted by computational algorithms such as mFOLD (*Zuker, 2003*). To incorporate chemical probing data, chemical reactivities are commonly converted to free energy folding constraints which are then used by RNA folding software (*Deigan et al., 2009*; *Mathews et al., 2004*; *Cordero et al., 2012*).

Notably, computational algorithms are poor at predicting tertiary structures in RNA, such as G-quadruplexes (*Vicens and Kieft, 2022*; *Sato and Kato, 2022*). The original predicted secondary structure of Spinach (*Paige et al., 2011*; *Strack et al., 2014*) was largely incorrect since it did not anticipate a G-quadruplex at the core of Spinach (*Fernandez-Millan et al., 2017*; *Cheng et al., 2017*). Addition of constraints from DMS MaP data does not improve modeling of G-quadruplexes since DMS MaP primarily identifies regions of canonical Watson-Crick base pairs (*Wells et al., 2000*). We therefore sought to apply BASH MaP to improve secondary structure predictions for RNA that contain a G-quadruplex.

To determine if BASH MaP can improve secondary structure prediction, we focused on Spinach. We first generated a secondary structure model from the Spinach crystal structure (PDB: 4TS2) as a benchmark for the comparison of future models (*Figure 5a*). Importantly, we only considered Watson-Crick interactions and non-complementary base pairs involving only two bases including G•A, A•A, and U•U base pairs when generating the secondary structure model (*Figure 5a*). We included these non-complementary base pairs because secondary structure modeling algorithms can model these interactions (*Do et al., 2006*). G-quadruplex, mixed tetrad, and base-triple interactions are intentionally left out as these tertiary structure elements are not predictable or able to be modeled by current secondary structure modeling algorithms (*Do et al., 2006*; *Reuter and Mathews, 2010*).

We next used a minimum free energy folding approach to generate a secondary structure model for Spinach (*Mathews et al., 1999*). We utilized mFold, a minimum free energy folding algorithm which attempts to find the base-pairing pattern with the lowest free energy for a given RNA sequence to predict the secondary structure of Spinach from sequence alone (*Zuker, 2003*; *Figure 5b*). The mFold secondary structure model of Spinach correctly displays stems P1 and most of P3 but lacks stem P2 and contains two stem loops which are not present in the crystallographic secondary structure (*Figure 5a–b*). The mFold secondary structure model demonstrates that minimum free energy folding algorithms fail to accurately model Spinach secondary structure from sequence alone.

We then asked whether incorporation of chemical probing data could improve the secondary structure prediction for Spinach. The most common method for using chemical probing data to improve

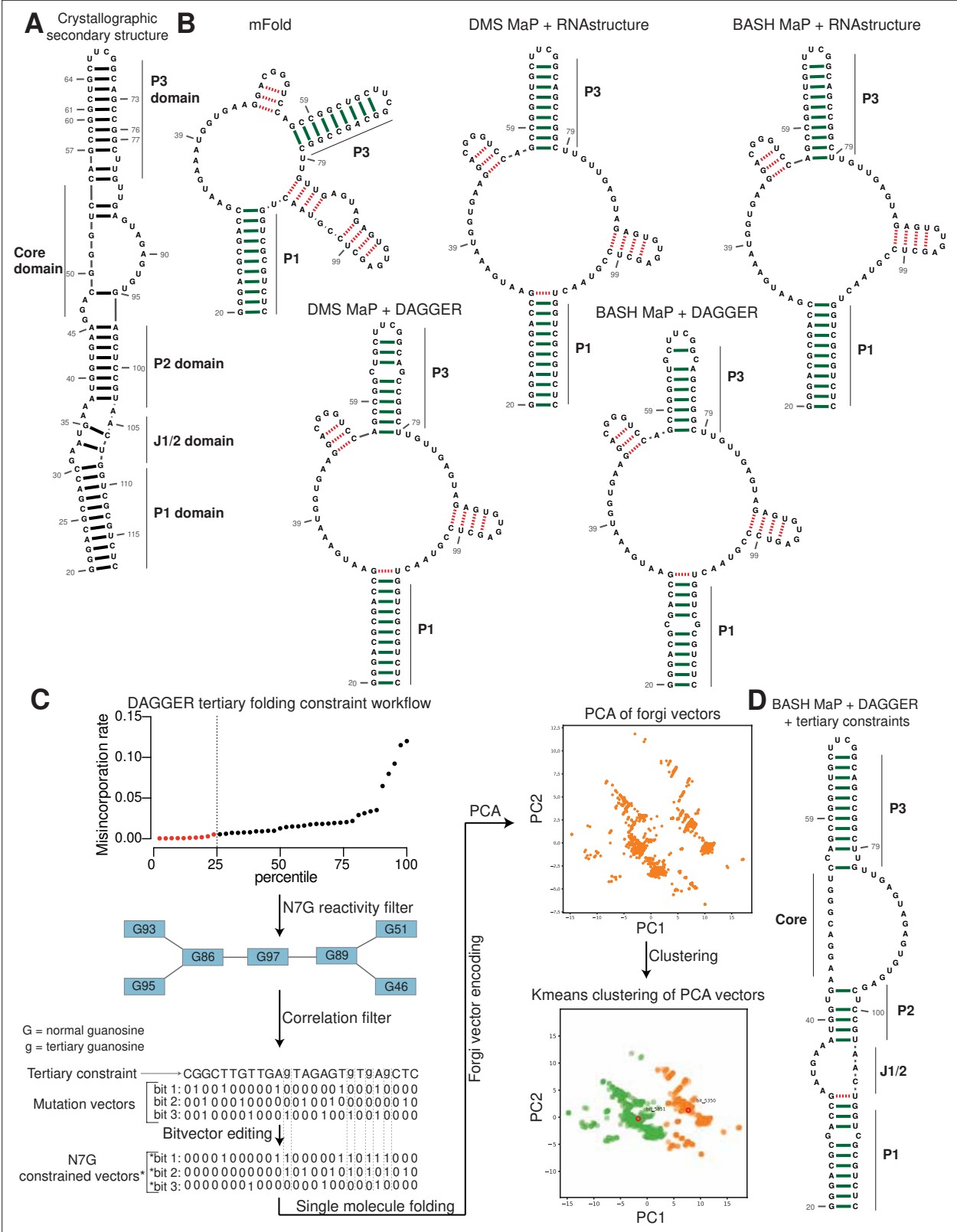

**Figure 5.** Tertiary folding constraints enable accurate secondary structure modeling of Spinach. (**A**) Spinach secondary structure model derived from the crystal structure 4TS2. This secondary structure model was utilized as a benchmark for comparing structure modeling approaches. (**B**) Spinach secondary structure models generated by the indicated combination of experimental data (DMS MaP or BASH MaP) and folding algorithm (mFold, RNAstructure, DAGGER). To determine whether structure probing data could improve the modeling of Spinach secondary structure, we assessed the

*Figure 5 continued on next page*

*Figure 5 continued*

sensitivity and specificity of a variety of computational approaches. Base pairs which are correctly predicted are indicated by green bars. Base pairs which are incorrectly predicted are indicated with red dashed lines. A base pair was determined to be correct if the true base pairing partner was within one base (+/-) from the indicated pairing partner. For detailed explanation of settings used for each RNA secondary structure modeling approach see Methods. Incorporation of experimental data improved Spinach secondary structure modeling; however, all structures included false helices and lacked the P2 domain of Spinach. (**C**) Tertiary-folding constraints derived from N7G-reactivity data are implemented through modification of the DaVinci data analysis pipeline (DAGGER). To generate tertiary constraints, G's in the bottom quartile of N7G reactivity are first identified. Then, all pairs of bottom quartile G's which display significant rates of co-occurring misincorporations with each other are identified as likely to be engaged in a tertiary interaction. These positions are indicated by annotating the base as lowercase in the input FASTA file for a modified DaVinci analysis pipeline. Each sequencing read is first converted to a bitvector where a zero represents no misincorporation and a one represents a misincorporation. The DaVinci pipeline forces sites of misincorporations to be single stranded upon subsequent folding. G's identified as likely to be engaged in a tertiary interaction are forced to be single stranded by editing the bitvectors and setting the value at each tertiary G to one. Sequencing reads with a misincorporation at any G identified as tertiary are treated separately because a modification at these positions indicates a change in tertiary structure. For these bitvectors, tertiary G's are allowed to be considered for base pairing by ContraFold, the folding engine used in DaVinci. After folding of each unique sequencing read, RNA secondary structures are converted to forgi vectors which utilize the Forgi library to encode RNA structure in a string of numbers. Principle component analysis (PCA) is then performed on the forgi vectors, and the ensemble of RNA structures is visualized through a plot of the first two principle components. Clustering of related RNA structures is performed in the PCA reduced dimensional space using techniques such as Kmeans clustering. Together, this pipeline enables more accurate structure modeling of G-quadruplex containing RNA. (**D**) Tertiary constraints and DAGGER analysis of BASH MaP-treated Spinach accurately model Spinach secondary structure. To determine whether tertiary folding constraints could improve Spinach structure modeling, we implemented the technique as described in (**C**) and applied it to Spinach BASH MaP data. Base pairs which are correctly predicted are indicated by green bars. Base pairs which are incorrectly predicted are indicated with red dashed lines. A base pair was determined to be correct if the true base pairing partner was within one base (+/-) from the indicated pairing partner. The resulting secondary structure most closely matches the crystallographic secondary structure through formation of the P2 domain and absence of false helices.

The online version of this article includes the following figure supplement(s) for figure 5:

**Figure supplement 1.** DAGGER clustering of Spinach BASH MaP.

secondary structure modeling is to convert chemical reactivities to free energy folding constraints which are then used by a minimum free energy folding program (*Mathews et al., 2004*). These constraints reward the algorithms for forming base pairs with RNA bases that display low chemical reactivity and penalize the formation of base pairs at locations with high chemical reactivity.

We first used DMS reactivities to generate free energy folding constraints for the folding program RNAstructure (*Rouskin et al., 2014*; *Reuter and Mathews, 2010*). We then compared the resulting secondary structure model to the mFold and crystallographic secondary structures (*Olson et al., 2022*; *Reuter and Mathews, 2010*; *Figure 5a–b*). For DMS MaP experiments, we considered chemical reactivities at all four nucleotides to create free energy folding constraints. For BASH MaP, we only considered chemical reactivities at A, C, and U bases to create free energy folding constraints as we previously showed the some based-paired G's displayed high N7G reactivity. If included, folding algorithms would incorrectly force base-paired G's with high N7G reactivities to be single stranded. Therefore, we ignored all data at G's when creating free energy folding constraints from BASH MaP data.

Incorporation of chemical reactivity data produced secondary structure models which retained the correct features of the mFold secondary structure including the P1 and P3 domain (*Figure 5b*). However, the secondary structure models continued to lack a properly formed P2 domain and continued to display two incorrect stem loops, although with less incorrect base pairs than the mFold model (*Figure 5b*). Together, these results show that free energy folding constraints from chemical probing data improve but are insufficient to correctly model the secondary structure of Spinach.

Next, we asked whether we could generate additional folding constraints from N7G reactivity data. We noticed that all the previous secondary structure models displayed the same incorrect stems (*Figure 5b*). These incorrect stems included G-quadruplex G's that were mis-assigned and forced into base pairs (*Figure 5b*). The low N1G reactivity of the G-quadruplex G's in DMS MaP was incorrectly interpreted by RNAstructure to indicate that G-quadruplex G's were engaged in a base-pair interaction. In principle, very low N7G reactivity could be used to identify G-quadruplex G's, which would then be restricted from being assigned as base paired.

We next implemented a two-step approach to confidently annotate G's engaged in a tertiary interaction (*Figure 5c*). First, we selected G's in the bottom quartile for N7G reactivity. Of these G's, we then identified which G's displayed high rates of co-occurring misincorporations with other lowly

reactive G's. Together, these two steps annotate G's engaged in a tertiary interaction and should thus be excluded from base pair assignments during secondary structure modeling.

To incorporate N7G reactivity data, we developed a single-molecule RNA secondary structure modeling pipeline called DAGGER (Deconvolution Assisted by N7G Gaps Enabled by Reduction and depurination). DAGGER modifies the recently described DaVinci analysis pipeline to incorporate N7G accessibility data derived from BASH MaP (*Yang et al., 2022*; *Figure 5c*). In contrast to minimum free energy folding algorithms like RNAstructure, DAGGER uses a thermodynamic independent RNA folding algorithm called CONTRAfold (*Do et al., 2006*). CONTRAfold utilizes a probabilistic methodology for generating secondary structures through statistical learning on large RNA secondary structure datasets (*Do et al., 2006*). A recent benchmark found that CONTRAfold outperformed thermodynamic folding algorithms in secondary structure modeling accuracy (*Wayment-Steele et al., 2022*).

DAGGER incorporates chemical reactivity data to constrain secondary structure modeling by CONTRAfold. Like DaVinci, DAGGER uses locations of misincorporations in a sequencing read to exclude those nucleotides from base-pair assignment by CONTRAfold (*Yang et al., 2022*). First, each sequencing read is converted to a string of ones and zeros called a bitvector which enables a simple representation of misincorporation sites within a sequencing read. A zero value indicates that nucleotide matches the reference sequence whereas a one indicates a misincorporation site (*Figure 5c*). DAGGER then folds each sequencing read as a unique molecule of RNA. Locations with a bitvector value of one are forced to be single stranded during secondary structure modeling by CONTRAfold (*Yang et al., 2022*). Thus, DAGGER utilizes chemical reactivity to constrain the secondary structure modeling of individual sequencing reads.

In contrast to chemical reactivity data on the Watson-Crick face, G's engaged in tertiary interactions display low reactivity and mostly appear as zeros in the DAGGER bitvectors. Consequently, CONTRAfold misinterprets G's engaged in tertiary interactions as engaged in secondary structure interactions which leads to inaccurate structure modeling (*Figure 5a–b*).

To prevent incorrect assignment of G's engaged in tertiary interactions, we created an additional step in DAGGER to edit each bitvector before folding. We edited each bitvector such that all G's engaged in tertiary interactions were represented by a one in the bitvector and would therefore be excluded from base-pair assignment (*Figure 5c*). We also set all G's not engaged in tertiary interactions to a bitvector value of zero as we previously determined N7G misincorporation status would incorrectly force base-paired G's to be single stranded (*Figure 2b*). Through N7G-directed editing of bitvector values, we incorporate tertiary-interaction constraints into the DAGGER pipeline.

We also wanted the DAGGER pipeline to be able to model single RNA molecules in distinct tertiary conformations. To accomplish this, we leveraged rare misincorporation events at G's engaged in tertiary interactions. In principle, a misincorporation at a G engaged in a tertiary interaction may result from an alternative conformation of the RNA in which the G is no longer engaged in a tertiary interaction. If a sequencing read contains a misincorporation at a G engaged in a tertiary interaction, we assume the sequencing read comes from an RNA in an alternative tertiary conformation. Furthermore, previously determined tertiary folding constraints may incorrectly prevent these G's from forming base pairs during structure modeling. Therefore, we differentially apply tertiary folding constraints to sequencing reads depending on whether they contain a misincorporation at a G engaged in a tertiary interaction. As such, our integration of N7G reactivity data into DAGGER allows us to model secondary structures for single RNA molecules in multiple distinct tertiary structures.

We next asked whether N7G-derived tertiary constraints improved the modeling of Spinach secondary structure. We implemented these tertiary-folding constraints through our structure probing deconvolution pipeline DAGGER which is based on the DaVinci pipeline (*Yang et al., 2022*; *Figure 5c*). Conventional analysis with DaVinci using either DMS MaP or BASH MaP Spinach datasets produced consensus secondary structures nearly identical to models obtained via free energy modeling approaches using RNAstructure (*Figure 5b*). We then applied the DAGGER pipeline to the 10,000 most modified sequencing reads in the Spinach BASH MaP dataset. Application of DAGGER to the BASH MaP Spinach dataset produced a consensus secondary structure that closely resembled the crystallographic secondary structure (*Figure 5d*). The resulting secondary structure model included a properly formed P1, P2, and P3 stem and was devoid of false helical regions (*Figure 5d*). We were unable to create tertiary constraints from DMS MaP since this method does not differentiate between

G-quadruplex G's and base-paired G's (*Figure 5—figure supplement 1c*). These results demonstrate that tertiary-folding constraints from BASH MaP as implemented through DAGGER enable accurate secondary structure modeling of Spinach.

## BASH MaP and DAGGER identifies a misfolded state of Spinach

Spinach has been used as a model RNA for studying RNA G-quadruplex folding (*Huang et al., 2014*; *Banco and Ferré-D'Amaré, 2021*). The fraction of fully folded Spinach transcripts was previously assessed using a fluorescence-based assay where the fluorescence of a solution of Spinach was compared to a standard solution containing a known amount of fully folded Spinach. Fluorescence-based assays showed that ~60% of Spinach was in a fluorescent form at 25 °C, with the remainder thought to be in a misfolded conformation (*Strack et al., 2013*). Preventing the misfolded conformation would increase overall Spinach fluorescence. Critically, it is unclear what, if any, structure is found in the misfolded form of Spinach. We therefore sought to apply BASH MaP and DAGGER to understand the conformation of Spinach misfolded states.

First, we sought to identify distinct conformations of Spinach using the RNA structure deconvolution program DANCE (*Olson et al., 2022*). DANCE uses a Bernoulli mixture model to identify mutually exclusive patterns of misincorporations on single sequencing reads from DMS MaP experiments. Sequencing reads that exhibit mutually exclusive patterns of misincorporations are derived from different conformations of RNA (*Homan et al., 2014*). We first used DANCE with the Spinach M2 DMS MaP dataset, which failed to identify more than one RNA conformation (data not shown). In contrast, when we used DANCE with the Spinach M2 BASH MaP dataset, we readily detected two states: State 1 with an abundance of 80.6% and State 2 with an abundance of 19.4% (*Figure 6a*). To confirm that the two states identified by DANCE were not a result of PCR-derived mutations, we repeated DANCE deconvolution of a Spinach BASH MaP dataset without the mutate-and-map step, which therefore uses RNA lacking PCR-derived mutations. DANCE deconvolution of this dataset also produced two states with abundances of 78.7% and 21.3% which closely resembles the abundances seen in the DANCE deconvolution of the M2 BASH MaP dataset (*Figure 6—figure supplement 1a*). Together, these data show that DANCE analysis of BASH MaP data uniquely identifies multiple conformations of Spinach.

We next asked whether the folding abundances measured by DANCE agreed with fluorescence-based measurements. The folding abundances identified by DANCE of ~80% are slightly higher than the ~60% folding abundance previously measured for Spinach (*Strack et al., 2013*). The slight difference in measured folding abundance may reflect additional conformations in Spinach which display similar chemical reactivity to the fluorescent-competent form of Spinach but are nonetheless unable to activate its fluorophore DFHBI-1T. DANCE deconvolution requires at least 10 nucleotides to display differential reactivities for deconvolution and would therefore be unable to differentiate between two conformations of Spinach which differ by only a few bases (*Olson et al., 2022*). Evidence for additional conformations within the DANCE identified states is indicated by intermediate reactivity levels in single stranded regions which should either display low or high reactivity (*Tomezsko et al., 2020*; *Figure 6a*). Overall, DANCE deconvolution of Spinach closely approximates fluorescence-based measurements of Spinach folding.

Next, we asked which regions of Spinach displayed different structure between the two states revealed by DANCE. Analysis of normalized misincorporation rate differences between State 1 and State 2 revealed large changes in the P2 and G-quadruplex core domains and minimal differences between stems P1, P3, and a linker region which was added to enable PCR amplification of Spinach (*Figure 6b*). The pattern of reactivity differences suggests that the 'misfolded' conformation of Spinach is folded and maintains key helical elements of the folded form of Spinach but has an alternative conformation of the P2 and G-quadruplex core domains.

Next, we wanted to understand the alternative conformation of the G-quadruplex. We first compared the reactivity to DMS for each G in both states (*Figure 6c*). G-quadruplex G's in State 1 displayed strong protections from DMS methylation (*Figure 6d*). In contrast, G-quadruplex G's in State 2 were found to display reactivity to DMS like base paired G's (*Figure 6e*). Stratification of G-quadruplex G's by either population average, State 1, or State 2 revealed increased N7G reactivity in State 2 (*Figure 6f*). Analysis of single stranded A's in the G-quadruplex loops showed an opposite trend with decreased reactivity to DMS in State 2 (*Figure 6g*). Together, these results suggest that

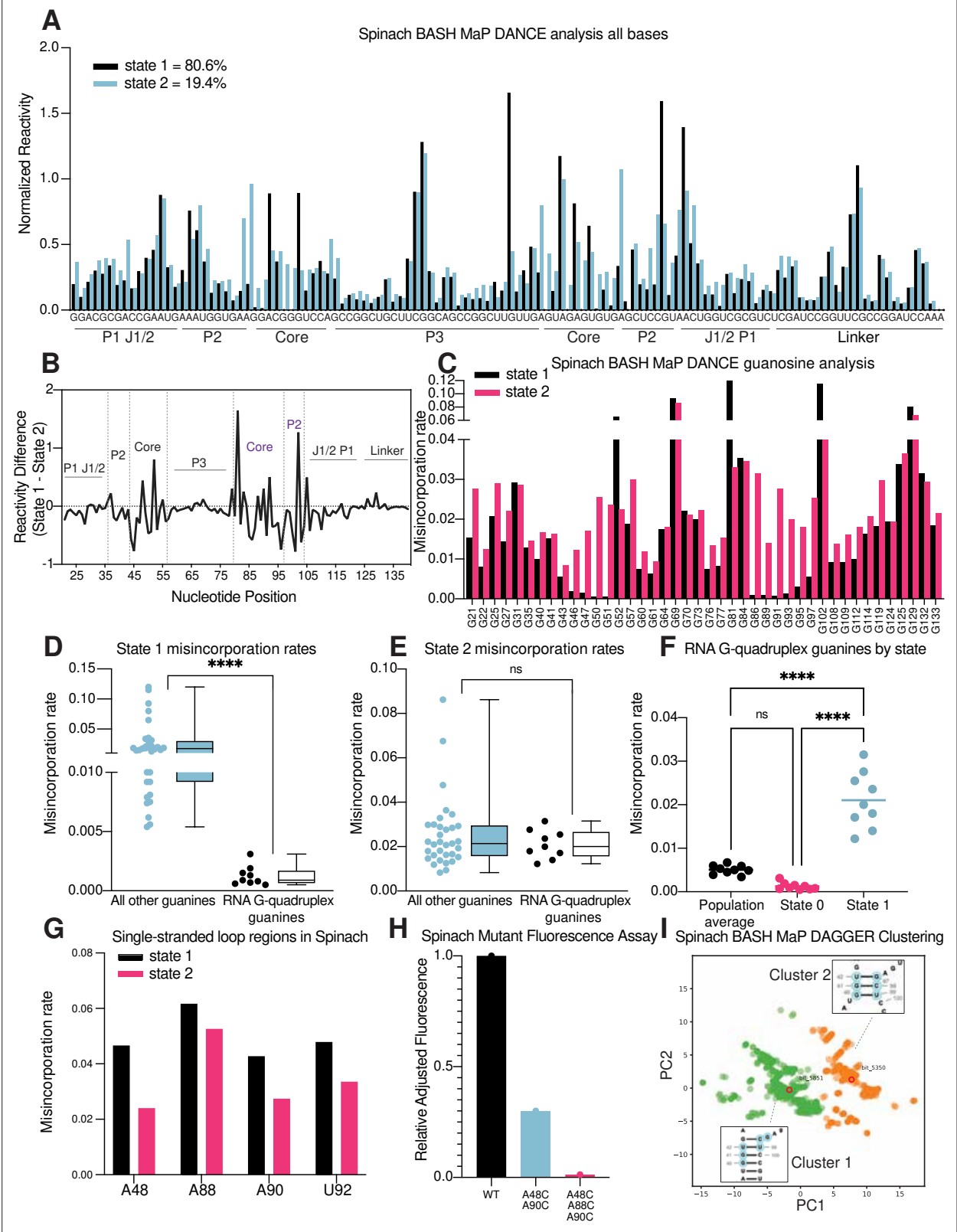

**Figure 6.** RNA structure deconvolution reveals G-quadruplex and P2 misfolding in Spinach. (**A**) RNA structure deconvolution identifies two conformations of Spinach. To identify multiple conformations of Spinach, we applied the program DANCE which utilizes a Bernoulli mixture model to identify mutually exclusive patterns of misincorporation in sequencing data. Spinach was subjected to BASH MaP and the sequencing data was input into the DANCE pipeline. DANCE identified two conformations denoted as State 1 and State 2 of 80.6% and 19.4% abundance. Misincorporation

*Figure 6 continued on next page*

*Figure 6 continued*

rates for each state were converted into reactivity values through normalization and plotted for comparison (see METHODS). (**B**) Reactivity differences between DANCE-identified states in Spinach reveal changes in G-quadruplex core and P2 domains. To identify which regions of Spinach adopted different structures between the two states identified by DANCE, we plotted the difference between reactivities of State 1 and State 2. The plot of change in reactivity shows that State 1 and State 2 differ in the core and P2 domains but remain unchanged in stems P1 and P3. (**C**) Spinach alternative states display differential N7G reactivity at G-quadruplex G's. To determine whether the alternative states of Spinach display differences in N7G reactivity, we compared the misincorporation rate of each G for State 1 and State 2. G-quadruplex G's are indicated below the plot in red. The plot shows that most G's in Spinach display no change in N7G reactivity to DMS whereas G-quadruplex G's show marked changes in N7G reactivity to DMS. (**D, E**) Spinach G-quadruplex G's are differentiated from all other G's in State 1 (**D**) ****p<0.0001, unpaired t-test with Welch's correction. G-quadruplex G's show no difference in misincorporation rates for State 2 (**E**) ns p=0.2262, unpaired t-test with Welch's correction. (**F**) The Spinach G-quadruplex is unfolded in State 2. To determine whether the G-quadruplex in Spinach was unfolded in State 2 we compared the misincorporation rate of G-quadruplex G's for the population average and DANCE deconvolved States 1 and 2. The plot shows that G-quadruplex G's display increased misincorporation rates in State 2 which suggests State 2 consists of an unfolded G-quadruplex. ****p<0.0001, Tukey's multiple comparison test. (**G**) Single-stranded loops in the Spinach G-quadruplex show decreased reactivity in State 2. To determine whether the single-stranded loops in the Spinach G-quadruplex core remain unpaired, we compared misincorporation rates between State 1 and State 2. The plot shows that all single-stranded loop residues show reduced reactivity to DMS in State 2 which suggests these positions display increased base-pairing interactions in State 2. (**H**) Nucleotide substitutions in Spinach G-quadruplex loop residues reduce Spinach fluorescence. To determine whether Spinach G-quadruplex loop residues make base-pair interactions with G-quadruplex G's in the Spinach alternative conformation, we systematically changed loop residues from A to C which should stabilize any base-pairing interactions with G's. We quantified Spinach fluorescence through an in-gel fluorescence assay (see Methods). The plot shows that conversion of A residues to C residues in the G-quadruplex loop region induces a progressive loss in Spinach fluorescence. (**I**) DAGGER clustering with tertiary constraints applied to Spinach BASH MaP data identifies two clusters with altered base pairing in the P2 domain. To identify alternative base pairing conformations of the misfolded Spinach, we utilized the orthogonal single molecule analysis method DaVinci with N7G reactivity data (DAGGER). We incorporated N7G reactivity data to create tertiary folding constraints before DAGGER folding and clustering (see Methods). Dimensional reduction and clustering identified two major clusters denoted Cluster 1 and Cluster 2. The most representative secondary structure of each cluster is boxed and indicated by the bit number. The DaVinci clustering plot reveals that the misfolded Spinach displays a register shift in the P2 domain. Cluster 1, colored green, is consistent with the Spinach crystallographic secondary structure. Cluster 2 is colored orange.

The online version of this article includes the following figure supplement(s) for figure 6:

**Figure supplement 1.** DANCE clustering of Spinach BASH MaP data and comparisons to Broccoli.

the misfolded Spinach adopts an alternative conformation characterized by a breakdown of N7G hydrogen bonding in the G-quadruplex core.

Next, we asked whether the misfolded Spinach contained alternative interactions of the G-quadruplex loop residues. The crystal structure of Spinach shows that the loop A's make no interactions within Spinach (*Huang et al., 2014*; *Warner et al., 2014*). Therefore, Spinach folding, and fluorescence is not expected to depend on the identity of the loop nucleotides. Since the loop A's showed decreased reactivity in State 2 and the G-quadruplex G's displayed similar reactivities to base-paired G's, we reasoned that the G-quadruplex G's may misfold via interactions with loop A's to form non-canonical G•A base pairs as seen in the P2 domain of Spinach (*Figure 6e* and *Figure 6g*).

If the misfolded Spinach contained non-canonical G•A base pairs between loop A's and G-quadruplex G's then mutation of loop A's to C should increase the stability of these interactions and therefore increase the fraction of misfolded Spinach transcripts. To increase the stability of these interactions, we progressively converted loop A's to C's and measured RNA fluorescence through an in-gel assay (*Filonov et al., 2015*). Surprisingly, mutation of two single stranded loop A's to C's resulted in ~40% of the wildtype fluorescence (*Figure 6h*). Mutation of three loop A's to C's resulted in near complete loss of fluorescence (*Figure 6h*). These results support a misfolded conformation of Spinach where G-quadruplex G's form non-canonical G•A base pairs with loop A's.

To better understand if the misfolded Spinach contains an alternative conformation in the P2 domain, we next asked whether an orthogonal RNA structure deconvolution technique, our previously developed DAGGER, produced clusters of RNA structure with alternative base pairing in the P2 domain. We previously developed DAGGER for modeling the secondary structure of Spinach through the incorporation of tertiary-folding constraints (see *Figure 5c–d*). DAGGER performs deconvolution after secondary structure modeling of individual RNA molecules (*Yang et al., 2022*), which contrasts with DANCE deconvolution which occurs before secondary structure modeling (*Olson et al., 2022*). To perform RNA structure deconvolution, DAGGER first folds each sequencing read utilizing misincorporations as folding constraints then represents each secondary structure as a string of numbers (*Thiel et al., 2019*). Then, principal component analysis (PCA) is used to reduce each string of numbers

representing RNA structures down to two unique values and plots these values as the first two principal components. From PCA plots, clustering methods such as Kmeans clustering can be used to identify clusters of similar RNA structures (*Yang et al., 2022*).

We analyzed the M2 BASH MaP dataset with tertiary constraints as implemented through the DAGGER pipeline and generated a plot of the first two principal components (*Figure 6i*). Interestingly, the DAGGER plot appeared to form two major clusters which agrees with the number of clusters identified by DANCE. To identify the most representative structure for each cluster, we performed Kmeans clustering with K equal to two. We then identified the sequencing read which was closest to the center of Cluster 1 as the most representative structure of Cluster 1. The same approach was used to identify the most represent structure of Cluster 2. We then compared the most representative secondary structures of Cluster 1 and Cluster 2. The key difference between the representative secondary structure of Cluster 1 and the representative secondary structure of Cluster 2 involved alternative base-pair interactions in the P2 domain (*Figure 5—figure supplement 1a–b*). The alternative base-pairing pattern of the P2 domain represents a register shift and is expected to prevent proper G-quadruplex folding. Together, DAGGER analysis supports an alternative base-pairing pattern in the P2 domain of Spinach's misfolded state.

We next asked whether Broccoli, a variant of Spinach identified through functional SELEX for better intracellular folding, contained stabilizing mutations in the P2 domain (*Filonov et al., 2014*). A previous alignment of the Broccoli sequence to Spinach suggested the two aptamers contain highly conserved G-quadruplex domains (*Filonov et al., 2014*). However, sequence differences in Spinach which led to the improved Broccoli aptamer are concentrated in the P2 domain (*Figure 6—figure supplement 1b*). In Spinach, a non-canonical A•A base pair just below the mixed tetrad was present as a G•C pair in Broccoli. Additionally, another non-canonical U•U base pair in Spinach was present as an A•U base pair in Broccoli. These sequence differences are expected to enhance folding of the fluorescent conformation by stabilizing the P2 domain. Furthermore, these mutations prevent the P2 domain from adopting the alternative Spinach base-pairing pattern seen in the secondary structure of the DaVinci Cluster 2 by converting a U42•G97 base pair to the unstable A42•G97 base pair (*Figure 5—figure supplement 1b*). Overall, these data provide a basis for the improved fluorescence of Broccoli and suggest that the misfolded state of Spinach involves a specific alternative base-pairing interaction of the P2 domain.

## Discussion

Deep sequencing of misincorporations induced by chemical probe-treated RNA is very useful for determining RNA structures and for identifying different RNA conformational states (*Siegfried et al., 2014*). Identification of different RNA conformational states relies on converting chemically modified nucleotides into a misincorporation during cDNA synthesis by a reverse transcriptase. As a result, only chemical adducts that are 'seen' by reverse transcriptases can be detected by cDNA sequencing. Adducts that occur on the Watson-Crick face perturb base pairing during nucleotide selection for cDNA synthesis, and therefore result in cDNA misincorporations. In the case of N7G, however, the methyl adduct is on the Hoogsteen face, rather than the Watson-Crick face and is thus largely invisible to reverse transcriptases. Importantly, information on the accessibility of N7G is needed to identify G-quadruplexes and any other tertiary interaction involving the N7G position. To detect N7G interactions, we developed BASH MaP. BASH MaP converts the otherwise nearly invisible $m^7G$ into a highly detectable modified nucleotide by selectively converting it to an abasic site. Using BASH MaP, we reveal the diversity of N7G interactions in RNA structure and enable precise localization of G-quadruplexes in RNAs in vitro and in cells. We then applied bioinformatic approaches to identify and measure changes in N7G interactions for previously intractable RNAs (*Figure 7*). Thus, BASH MaP expands the structures that can be detected using DMS-based structure probing methods.

G-quadruplexes have diverse roles in regulating RNA function but have been challenging to identify and study in vitro and in cells (*Arora et al., 2008*; *Marcel et al., 2011*; *Blice-Baum and Mihailescu, 2014*; *Zhang et al., 2019b*; *Asamitsu et al., 2023*; *Kharel et al., 2023*). The lack of methods to specifically identify clusters of the interconnected guanosine residues that constitute a G-quadruplex makes it difficult to validate and establish the topology of G-quadruplexes in RNA. This limits the ability to establish the physiologic roles and functions of G-quadruplexes. Current RNA structure mapping methods cannot detect G-quadruplexes and other complex RNA structures because they

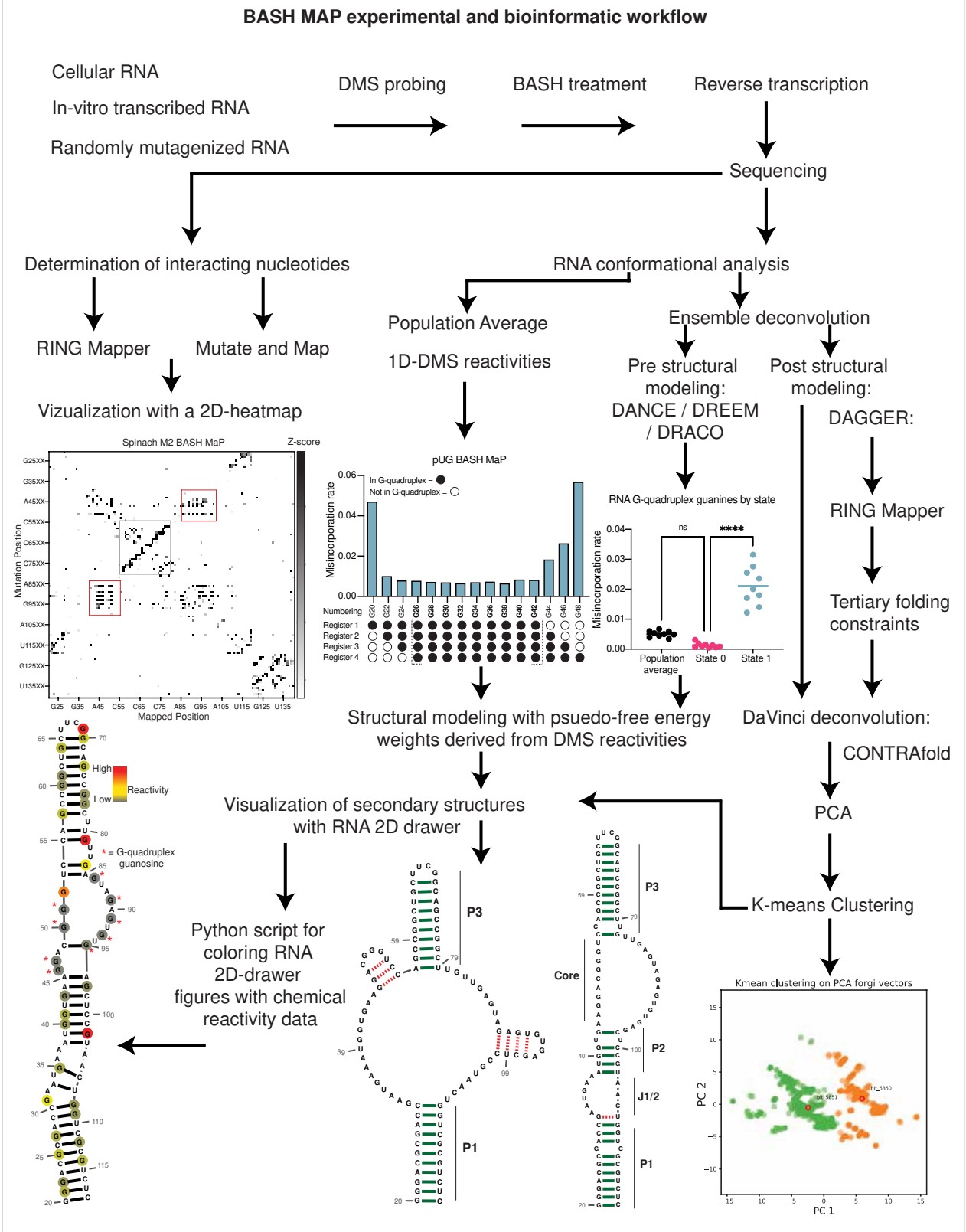

**Figure 7.** Summary BASH MaP experimental and bioinformatic workflow. Overview of the BASH MaP experimental and bioinformatic workflow.

The online version of this article includes the following figure supplement(s) for figure 7:

**Figure supplement 1.** Validation of minimum separation between misincorporations for SuperScript II and Marathon RT.

*Figure 7 continued on next page*

*Figure 7 continued*

**Figure supplement 2.** Discrimination between multi-hit versus single-hit mechanisms for co-occurring misincorporations between G's in Spinach BASH MaP data.

primarily detect the presence of Watson-Crick base pairs. A feature of BASH MaP is its ability to specifically identify G's involved in a G-quadruplex.

G-quadruplex G's can be identified based on their very low reactivity towards DMS, as measured by a low misincorporation rate. However, G-quadruplex G's are more precisely identified through co-occurring misincorporations in single reads where individual G's in a G-quadruplex become methylated with DMS, which disrupts the G-quadruplex, resulting in coordinated methylation of the otherwise non-reactive G's in the G-quadruplex. These clusters of reactive G's will occur on single reads, reflecting individual strands of RNA. Notably, the heatmaps used to visualize BASH MaP analyses of RNA structure produce a new box-like pattern for G-quadruplexes not previously seen with DMS MaP heatmaps. The statistical co-occurrence of misincorporations at G's in BASH MaP allows us to create a network of structurally interconnected G's, which thus identifies G-quadruplexes.

The G's involved in a G-quadruplex can similarly be identified using the mutate-and-map method, in which mutations are intentionally incorporated into an RNA using mutagenic PCR (*Cheng et al., 2017*). In this method, PCR-derived mutation of any G in the G-quadruplex enhances the reactivity of all other G's in the G-quadruplex, as well as another structural feature that is dependent on G-quadruplex folding, as seen in Spinach. Overall, the M2 BASH MaP method allows highly precise identification of specific G's involved in G-quadruplex formation.

BASH MaP provides substantial improvements over previous methods for discovering G-quadruplex G's in vitro. G-quadruplexes G's have been predominantly identified by performing chemical or enzymatic probing of an RNA refolded in potassium-containing buffer versus refolding in lithium- or sodium-containing buffer (*Su et al., 2014*). Since G-quadruplexes are destabilized by lithium or sodium ions, refolding an RNA in lithium buffer or sodium buffers should lead to reactivity changes around G-quadruplexes. Similarly, chemical or enzymatic reactivity of RNA synthesized with 7-deaza GTP, which is unable to form G-quadruplexes, can be compared with the reactivity of RNA synthesized with GTP to identify regions where G-quadruplexes may form (*Weldon et al., 2017*). However, these methods are limited to cell-free RNA and only infer locations of G-quadruplexes. BASH MaP enables high-throughput identification of G-quadruplexes G's in an RNA of interest by directly detecting N7G accessibility.

BASH MaP also provides substantial improvements over previous methods for identifying G-quadruplex G's in cells. Chemical probes such as kethoxal (*Weng et al., 2020*), which modify the Watson-Crick face of G, lack the ability to discriminate between G's in helices and G-quadruplexes (*Figure 2—figure supplement 1c*). Other assays rely on reverse transcriptase stalls at G-quadruplexes, but these methods only indicate the 3' most guanine and do not definitively identify a G-quadruplex as the source of the stall (*Guo and Bartel, 2016*; *Kwok et al., 2016a*; *Kharel et al., 2023*). SHAPE reagents were found to preferentially modify terminal G's in some G-quadruplexes; however, this property does not have a clear mechanistic basis and does not unambiguously identify G-quadruplexes G's (*Kwok et al., 2016b*). BASH MaP identifies G-quadruplexes G's in cells with RING MaP co-occurring misincorporation data by directly revealing G-quadruplex guanosine hydrogen bonding networks.

BASH MaP also enables prediction of G-quadruplex conformations in cells. Our analysis identified a unique G-quadruplex formation in the *AKT2* 3'UTR, characterized by an atypical topology that includes a significantly elongated internal loop. Indeed, recent studies reported the in vitro formation of RNA G-quadruplexes with unusually long loop lengths in human 5'UTRs (*Jodoin et al., 2014*). However, current computational algorithms cannot identify which guanines form the G-quadruplex in atypical or long-loop G-quadruplexes (*Mukundan and Phan, 2013*). BASH MaP simplifies this process by identifying long-range G – G correlations, thus enabling predictions for the conformations of the *AKT2* G-quadruplex.

Although BASH MaP identifies G's engaged in G-quadruplexes, the precise topology of a G-quadruplex remains obscured in both RING MaP and mutate-and-map heatmaps. Unlike traditional mutate-and-map experiments which directly reveal base-paired positions, mutation of a single G within the G-quadruplexes of Spinach and pUG leads to increases in N7G reactivity for G's in multiple

quartets (*Figure 4D* and *Figure 4—figure supplement 1b*). We attribute this observation to a disruption of base-stacking within the G-quadruplex from DMS modification or an alternative conformation which then causes global destabilization of the G-quadruplex fold. Together, this data suggests that G-quadruplexes are uniquely sensitive to precise nucleotide sequences and may adopt an alternative structure upon perturbation of even a single G residue.

An important advance in RNA structure mapping was the utilization of multiple modification events on a single RNA molecule to detect the presence of different conformations of RNA in solution (*Homan et al., 2014*; *Olson et al., 2022*; *Yang et al., 2022*; *Tomezsko et al., 2020*; *Morandi et al., 2021*). The power of deconvolution methods increases with the number of co-occurring misincorporations on individual reads by identifying additional mutually exclusive patterns of nucleotide accessibility. Mutually exclusive patterns indicate different structures. In BASH MaP, $m^7G$ contributes to the misincorporations seen in a read. Since N7G is the most reactive site to DMS, the detection of $m^7G$ results in substantially more misincorporations per read than in conventional DMS MaP. As a result, the ability to identify distinct conformational states of RNA is substantially improved by detecting the $m^7G$ formed in DMS experiments.

We used BASH MaP to understand the distinct conformations of Spinach. Spinach fluorescence is limited since it exists in both a folded and a non-fluorescent unfolded conformation (*Strack et al., 2013*). We first developed a computational approach for utilizing BASH MaP data to create new constraints for RNA folding. Our DAGGER pipeline uses a mixture of chemical reactivity data and single-molecule co-occurring misincorporation data to identify G's likely to be engaged in a tertiary interaction. DAGGER then restricts identified G's from forming canonical base pairs during folding. The incorporation of tertiary folding constraints through DAGGER led to a remarkably accurate prediction of the Spinach secondary structure (*Figure 5d*).

Surprisingly, DANCE analysis of Spinach BASH MaP data revealed that the previously described 'misfolded' conformation was instead a highly specific folded conformation. In the non-fluorescent conformation, the G-quadruplex G's only display a modest increase in reactivity, and stems P1 and P3 remained intact (*Figure 6b–c*). Instead, the G-quadruplex G's display N7 reactivity consistent with G's engaged in base-pair interactions (*Figure 6e*). In addition, the hyper-reactive G81 undergoes a dramatic decrease in reactivity in the non-fluorescent conformation which indicates a change in local base stacking. The decrease of G81 reactivity suggests that G81 is no longer located at the end of a helix in the nonfluorescent conformation (*Figure 6c*). Also, DAGGER analysis supports a reorganization the of P2 domain which is likely aided by the adjacent and flexible J1/2 domain of Spinach (*Figure 6h*). Together, these results suggest that the non-fluorescent conformation of Spinach involves the G-quadruplex G's forming an extended P3 domain characterized by base-pairing with loop residues and a reorganization of the P2 domain.

BASH MaP also revealed insights into how Spinach fluorescence is enhanced by sequence alterations. Notably, Broccoli, an RNA aptamer evolved and selected for higher fluorescence, shows high sequence similarity to Spinach except for key mutations in the P2 stem and residues proximal to the G-quadruplex core (*Filonov et al., 2014*; *Figure 6—figure supplement 1b*). The mutations of Broccoli are in locations which do not impact G-quadruplex folding, but instead stabilize the P2 domain, and destabilize the alternative base pairing pattern of the P2 domain (*Figure 5—figure supplement 1b* and *Figure 6—figure supplement 1b*). In this way, Broccoli enhances the fluorescence of the Spinach aptamer by destabilizing the non-fluorescent alternative tertiary conformation (*Filonov et al., 2014*).

## Methods
### Reagents and equipment

All buffers and NTPs were purchased from commercial sources. DMS was purchased from Sigma-Aldrich (77-78-1). Potassium borohydride was purchased from Santa Cruz Biotechnology (SC-250747). SuperScript II reverse transcriptase was purchased from Invitrogen. Marathon RT was purchased from Kerafast. DFHBI-1T was purchased from Lucerna Technologes. N-methyl mesoporphyrin IX (NMM) was purchased from Santa Cruz Biotechnology. DNAse I was purchased from Sigma-Aldrich. DNA constructs and primers were ordered from Integrated DNA Technologies (IDT) and Twist Bioscience (*Table 1* and *Table 2*). Spin columns were purchased from Zymo Research. AMPure XP beads were

**Table 1.** Step 1 PCR primers.

| RNA sample name | Step 1 forward primer sequence | Step 1 reverse primer sequence |
|---|---|---|
| SSII_HeLa_18 s_0 min_1 | ACACGACGCTCTTCCGATCTNNNNN**ATAT**CCCGTTGAACCCCATTCGTGA | GACGTGTGCTCTTCCGATCTNNNNN<u>TAAGGCG</u>AGTGTGTACAAAGGGCAGGGAC |
| SSII_HeLa_18 s_0 min_2 | ACACGACGCTCTTCCGATCTNNNNN**CGCG**CCCGTTGAACCCCATTCGTGA | GACGTGTGCTCTTCCGATCTNNNNN<u>TAAGGCG</u>AGTGTGTACAAAGGGCAGGGAC |
| SSII_HeLa_18 s_30 min_1 | ACACGACGCTCTTCCGATCTNNNNN**ATAT**CCCGTTGAACCCCATTCGTGA | GACGTGTGCTCTTCCGATCTNNNNN<u>CGTACTAG</u>GTGTGTACAAAGGGCAGGGAC |
| SSII_HeLa_18 s_30 min_2 | ACACGACGCTCTTCCGATCTNNNNN**CGCG**CCCGTTGAACCCCATTCGTGA | GACGTGTGCTCTTCCGATCTNNNNN<u>CGTACTAG</u>GTGTGTACAAAGGGCAGGGAC |
| SSII_HeLa_18 s_1 hour_1 | ACACGACGCTCTTCCGATCTNNNNN**ATAT**CCCGTTGAACCCCATTCGTGA | GACGTGTGCTCTTCCGATCTNNNNN<u>AGGCAGAA</u>GTGTGTACAAAGGGCAGGGAC |
| SSII_HeLa_18 s_1 hour_2 | ACACGACGCTCTTCCGATCTNNNNN**CGCG**CCCGTTGAACCCCATTCGTGA | GACGTGTGCTCTTCCGATCTNNNNN<u>AGGCAGAA</u>GTGTGTACAAAGGGCAGGGAC |
| SSII_HeLa_18 s_2 hour_1 | ACACGACGCTCTTCCGATCTNNNNN**ATAT**CCCGTTGAACCCCATTCGTGA | GACGTGTGCTCTTCCGATCTNNNNN<u>TCCTGAGC</u>GTGTGTACAAAGGGCAGGGAC |
| SSII_HeLa_18 s_2 hour_2 | ACACGACGCTCTTCCGATCTNNNNN**CGCG**CCCGTTGAACCCCATTCGTGA | GACGTGTGCTCTTCCGATCTNNNNN<u>TCCTGAGC</u>GTGTGTACAAAGGGCAGGGAC |
| SSII_HeLa_18 s_3 hour_1 | ACACGACGCTCTTCCGATCTNNNNN**ATAT**CCCGTTGAACCCCATTCGTGA | GACGTGTGCTCTTCCGATCTNNNNN<u>GGACTCCT</u>GTGTGTACAAAGGGCAGGGAC |
| SSII_HeLa_18 s_3 hour_2 | ACACGACGCTCTTCCGATCTNNNNN**CGCG**CCCGTTGAACCCCATTCGTGA | GACGTGTGCTCTTCCGATCTNNNNN<u>GGACTCCT</u>GTGTGTACAAAGGGCAGGGAC |
| SSII_HeLa_18 s_4 hour_1 | ACACGACGCTCTTCCGATCTNNNNN**ATAT**CCCGTTGAACCCCATTCGTGA | GACGTGTGCTCTTCCGATCTNNNNN<u>TAGGCATG</u>GTGTGTACAAAGGGCAGGGAC |
| SSII_HeLa_18 s_4 hour_2 | ACACGACGCTCTTCCGATCTNNNNN**CGCG**CCCGTTGAACCCCATTCGTGA | GACGTGTGCTCTTCCGATCTNNNNN<u>TAGGCATG</u>GTGTGTACAAAGGGCAGGGAC |
| MaRT_HeLa_18 s_4 hour_1 | ACACGACGCTCTTCCGATCTNNNNN**ATAT**CCCGTTGAACCCCATTCGTGA | GACGTGTGCTCTTCCGATCTNNNNN<u>TAGGCATG</u>GTGTGTACAAAGGGCAGGGACCATGCCTA |
| MaRT_HeLa_18 s_4 hour_2 | ACACGACGCTCTTCCGATCTNNNNN**CGCG**CCCGTTGAACCCCATTCGTGA | GACGTGTGCTCTTCCGATCTNNNNN<u>TAGGCATG</u>GTGTGTACAAAGGGCAGGGACCATGCCTA |
| Structure Cassette Universal | ACACGACGCTCTTCCGATCTNNNNNGGCTGGCCTTTCGGGCCAA | GACGTGTGCTCTTCCGATCTNNNNNGAACCGGACCGAAGCCCG |
| AKT2 3UTR | ACACGACGCTCTTCCGATCTNNNNNAACACCTCTGGGTGTTTGGAGTTTAGC | GACGTGTGCTCTTCCGATCTNNNNNCCGTACAAATATGAAGACGAGGAGAAAGGC |

**Table 2.** Oligos and primers.

| Oligos | Primers |
|---|---|
| Spinach oligo | GTATAATACGACTCACTATAGGGCTGGCCTTTCGGGCCAA GGGACGCGACCGAATGAAATGGTGAAGGACGGGTCCAG CCGGCTGCTTCGGCAGCCGGCTTGTTGAGTAGAGTGTGAG CTCCGTAACTGGTCGCGTCTCGATCCGGTTCGCCG GATCCAAATCGGGCTTCGGTCCGGTTC |
| pUG oligo | GGCTGGCCTTTCGGGCCAAGTGTGTGTGTGTGTGTGTGTGTGTGTTCGATCCGGTTCGCCGGATCCAAAT |
| Structure Cassette T7 Forward | GTATAATACGACTCACTATAGGGCTGGCCTTTCGGGCCAA |
| Structure Cassette T7 Reverse | GAACCGGACCGAAGCCCGATTTGGATCCGGCGAACCGGAT |
| Structure Cassette RT primer | GAACCGGACCGAAGCCCGA |
| Spinach A48C, A90C template | ACGGGCCAGATATACGCGTAGTTCCTGCTATAATTAGCCTTCCTCATAAGTTGCACTGCTCCAGGTGATAGTGCGGGAACCTCGATGGTCTTCA CACTTTACTTCAGCGTCtggtaggcgtgtacggtgggaggcctatataagcagagctTCTGGCTAACTAGGCTGGCCTTTCGGGCCAAGGGACGCGACCGAATG AAATGGTGAAGGCCGGGTCCAGCCGGCTGCTTCGGCAGCCGGCTTGTTGAGTAGCGTGTGAGCTCCGTAACTGGTCGCGTCTCGATCCGG TTCGCCGGATCCAAAT |
| Spinach A48C, A88C, A90C template | ACGGGCCAGATATACGCGTAGTTCCTGCTATAATTAGCCTTCCTCATAAGTTGCACTGCTCCAGGTGATAGTGCGGGAACCTCGATGGTCTTCA CACTTTACTTCAGCGTCtggtaggcgtgtacggtgggaggcctatataagcagagctTCTGGCTAACTAGGCTGGCCTTTCGGGCCAAGGGACGCGACCGAATG AAATGGTGAAGGCCGGGTCCAGCCGGCTGCTTCGGCAGCCGGCTTGTTGAGTCGCGTGTGAGCTCCGTAACTGGTCGCGTCTCGATCCGGTT CGCCGGATCCAAAT |

purchased from Beckman Coulter. Barcoded sequencing oligos were purchased from New England BioLabs. AmpliScribe T7 High Yield Transcription Kit were purchased from Biosearch Technologies.

## Cell culture

HeLa cells were grown in DMEM containing 10% FBS with 100 U ml$^{-1}$ penicillin and 100 µg ml$^{-1}$ streptomycin under standard culturing conditions (37 °C with 5.0% $CO_2$). SH-SY5Y cells were grown in a 1:1 ratio of DMEM to F12 media containing 10% FBS with 100 U ml$^{-1}$ penicillin and 100 µg ml$^{-1}$ streptomycin under standard culturing conditions (37 °C with 5.0% $CO_2$).

## Reduction and depurination procedure (BASH MaP treatment)

Reduction and depurination procedures were adapted from a previously published protocol (*Zhang et al., 2022*). Between 200 ng and 2 µg of DNAse-I-treated RNA was brought to 10 µL in water and added to 40 µL of freshy prepared 1 M Potassium Borohydride in RNAse-free water (final reaction concentration 800 mM). The reaction was mixed and placed in the dark at room temperature for various lengths of time. To stop the reaction, 4 volumes of Zymo RNA bindng buffer (200 µL) was slowly added to avoid foaming. An equal volume of 100% ethanol (250 µL) was then added to precipitate the RNA. RNA was recovered with RNA clean and concentrator 5 columns and eluted with 45 µL of RNAase free water. Eluted RNA (45 µL) was added to 5 µL of 1 M sodium acetate / acetic acid buffer (pH 2.9) and incubated in the dark for 4 hours at 45 °C. Reactions were purified with Oligo clean and concentrator kits following manufacturer's instructions and eluted in 15 µL water.

## HeLa 18s rRNA reduction optimization

HeLa cells were grown to 80% confluence in 75 mm flasks. Cells were pelleted by centrifugation at 400 x *g* for 3 min followed by 1 wash with PBS. Cells were then lysed by the addition of 4 mL TRIzol reagent. Total RNA was isolated following manufacturer's instructions. 25 µg of total RNA was subjected to DNAse I treatment and purification with Zymo RNA Clean and Concentrator 25 columns according to manufacturer's instructions. 1.8 µg of DNAse-I-treated HeLa total RNA was input for reduction and depurination and random hexamers were used for priming the reverse transcription reaction.

## Reverse transcription

Superscript II: Reverse transcription with SSII was performed as previously described (; *Mitchell et al., 2023*; *Siegfried et al., 2014*). Briefly, 200 ng of input RNA was mixed with 1 µL 50 µM random hexamer (HeLa total RNA) or 2 µL of 1 µM gene specific reverse transcription primer (in vitro transcribed RNAs) and annealed. Annealed RNA was added to reverse transcription master mix containing 6 mM final concentration of $MnCl_2$. Reverse transcription reactions were incubated at 23 °C for 15 min only when using random hexamers then 42 °C for 3 hours.

Marathon RT: Reverse transcription with Marathon RT was performed with a modified protocol. First, 200 ng of input RNA was mixed with 2 µL10 mM dNTP and 1 µL 50 µM random hexamer or 2 µL of 1 µM gene-specific RT primer. The volume was brough up to 8 µL and the RNA was annealed to the primer with the following thermocycler settings: 80 °C for 1 min, 65 °C for 5 min at which point the RNA was placed on ice for 2 min. Then, an RT master mix was created containing 4 µL of 5 x Marathon MaP buffer (250 mM Tris-HCl pH 7.5, 1 M NaCl), 4µLof 100% glycerol, 1 µL of 100 mM DTT, 0.5 µL of 20 mM $MnCl_2$, 0.5 µL of water. Next, 10 µL of RT master mix was added to 8 µL of annealed RNA followed by the addition of 2 µL Marathon RT. Reverse transcription reactions were incubated at 23 °C for 15 min only when using random hexamers then 42 °C for 3 hr.

Reverse transcription reactions were inactivated by incubating at 75 °C for 10 min. cDNA was purified with DNA clean and concentrator 5 columns by using 7 volumes of DNA binding buffer (140 µL) and eluted in 10 µL water.

## Sequencing library preparation

Sequencing libraries were generated via a two-step PCR. Briefly, one-fifth of purified cDNA (2 µL) was amplified with step 1 forward and reverse primers. HeLa 18 s rRNA samples included a 4-nucleotide barcode on the forward primer and a 7-nucleotide barcode on the reverse primer which enabled pooling of purified step 1 PCR products. AKT2 step 1 PCR primers were designed to amplify a ~200

nucleotide segment of the *AKT2* 3'UTR. In vitro transcribed RNAs used non-barcoded step 1 forward and reverse primers which were specific to flanking sequences introduced into the RNA constructs. These flanking sequences are referred to as a Structure Cassette (SC). Step 1 PCR reactions were assembled on ice by adding to 2 µL purified cDNA: 1.25 µL of 10 µM forward primer, 1.25 µL of 10 µM reverse primer, 9 µL of water and 12.5 µL of 2 x Phusion HF master mix or 2 x Phusion GC master mix for AKT2 step 1 PCR. Step 1 PCR reactions were run with the following thermocycler settings: 98 °C for 2 min followed by 8–28 cycles of 98 °C for 10 s, 65 °C for 20 s, 72 °C for 20 s, followed by a final extension of 2 min. PCR cycle number was optimized for each experiment separately. PCR reactions were purified with 1.8 x AMPure XP beads according to manufactures instructions. Purified DNA was eluted in 25 µL water.

Step 2 PCR was performed with NEB Next Multiplex Oligos for Illumina. Step 2 PCR reactions were composed of the following: 2–10 ng of purified step 1 DNA brough up to 12 µL water, 4 µL of i5 index primer or universal primer, 4 µL i7 index primer, and 20 µL of 2 x Phusion HF or 2 x Phusion GC for AKT2 library prep. Step 2 PCR reactions were run with the following thermocycler settings: 98 °C for 2 min followed by 6–12 cycles of 98 °C for 15 s, 65 °C for 30 s, 72 °C for 30 s, then 72 °C for 5 min. PCR cycle number was optimized for each experiment separately. Step 2 PCR reactions were size selected with AMPure XP beads, eluted in 25 µL of water, and quantified with Qubit 1 x dsDNA HS kit. Libraries were pooled and sequenced on the NovaSeq 6000 PE 2x100, NovaSeq 6000 PE 2x150, MiSeq PE 2x150, Nextseq2000 P1 600 cycles, or MiSeq Micro 300 cycles.

## RNA construct preparation

RNA constructs were designed to include flanking stem loops on the 5' and 3' ends of the RNA that are designed to fold back on themselves as not to interfere with the folding of the RNA of interest. When present, these flanking sequences are referred to as a structure cassette (SC). RNAs of interest were ordered from IDT or Twist and then PCR amplified to introduce a T7 promoter as well as the 5' and 3' structure cassette sequences. PCR reactions were purified with 1.8 X AMPure XP beads. RNA was in vitro transcribed with ~200 ng of template dsDNA with AmpliScribe T7 High Yield Transcription Kit following manufacturer's instructions. Reactions were purified with RNA Clean and Concentrator 25 kits. RNA quality was assessed on a 10% denaturing polyacrylamide gel and stained with SYBR gold.

## DMS modification of in vitro transcribed RNA

In vitro transcribed RNA was probed as previously described (*Siegfried et al., 2014*). Briefly, 200 ng – 1 µg of RNA was folded in a buffer designed to promote modification of A and C bases only (200 mM Bicine pH 7.75 at room temperature, 100 mM KCl) or a buffer which promotes the additional modification of $m^1G$ and $m^3U$ (200 mM Bicine pH 8.37 at room temperature, 100 mM KCl). RNA was refolded in buffer by heating to 95 °C for 3 min then snap cooling on ice for 1 min. Then $MgCl_2$ was added to a final concentration of 1–5 mM and the RNA was left at room temperature for 10 min. Neat DMS was diluted to 1.7 M with anhydrous ethanol and added to the folded RNA for a final DMS concentration of 170 mM. DMS reactions was performed for 10 min at room temperature or 6 min at 37 °C. Spinach was probed at room temperature due to its thermal instability. pUG RNA was probed at 37 °C. Reactions were quenched by the addition of 4 volumes of 20% beta-mercaptoethanol in water on ice. RNA was purified from the quenched DMS reaction with RNA Clean and Concentrator 5 kit and eluted in 10 µL water. RNA was either directly processed or stored at –80 °C.

## In cell DMS modification and purification

DMS modification of SH-SY5Y cells was performed as previously described (*Olson et al., 2022*). Briefly, cells were grown to 85% confluence in a 10 cm dish. Then, the media was exchanged for a DMS probing media which consisted of 1:1 DMEM to F12 medium with 10% FBS and 200 mM Bicine adjusted to pH 8.37 with NaOH at room temperature. Probing media was prewarmed to 37 °C and cell media was exchanged with 5.4 mL probing media for 3 min at 37 °C. Then 600 µL of prewarmed 1.7 M DMS or prewarmed 600 µL ethanol was added to the cells and incubated for 6 min at 37 °C. DMS reaction was stopped by the addition of 6 mL 20% 2-mercaptoethanol and cells were placed on ice. We noticed that DMS probing caused a large portion of adherent cells to become detached from the plate. We therefore decanted the quenched DMS probing media off the plate into a 15 mL falcon

tube and pelleted the detached cells for 5 min at 4 °C and 1000 x *g*. To isolate the total RNA, 2 mL Trizol reagent was added to the plate for the cells which had not become detached. Then, the 2 mL Trizol reagent was transferred to solubilize the pelleted cells which had become detached during DMS probing. Total RNA was then isolated according to manufacturer's instructions.

## Gene-specific BASH MaP

Total RNA was treated with DNAse I for 30 min prior to BASH MaP treatment according to manufacturer's instructions. Three replicates of 1 µg of total RNA was mixed with 1 µL 50 µM random hexamers and the volume was adjusted to 11 µL. RNA was annealed by heating to 85 °C for 1 min, 65 °C for 5 min, then placing on ice for 3 min. Reverse transcription was performed with SuperScript II as described previously (*Cheng et al., 2017*). Reverse transcription reactions were incubated at 23 °C for 15 min and then 42 °C for 3 hours. Reactions were inactivated by heating to 75 °C for 10 min. Reverse transcription replicates were pooled and the cDNA was purified with the Zymo DNA Clean and Concentrator 5 kit and eluted in 10 µL water.

## BASH MaP data processing

Sequencing data was processed using the ShapeMapper pipeline (*Mitchell et al., 2023*; *Busan and Weeks, 2018*).

For HeLa 18 s rRNA experiments, sequencing data was aligned to 12 unique 18 S amplicons defined by the unique combinations of 4-nucleotide forward and 7-nucleotide reverse barcode sequences inserted during step 1 PCR. Sequencing data alignment was performed with ShapeMapperV2.2 with the following settings: `--amplicon --output-parsed --output-aligned-reads --nproc 15 --output-counted-mutations --min-mutation-separation 0`.

Optimal misincorporation separation distance for DMS MaP experiments was identified by analyzing the misincorporation signatures of $m^1G$ and $m^3$psuedoU in *E. coli* rRNA as surrogates for $m^1A$ and $m^3C$ (*Figure 7—figure supplement 1a–b*). *E. coli* 23 s rRNA contains two bases methylated on their Watson-Crick face, $m^1G745$ and $m^3$pseudoU1915. To analyze whether reverse transcriptases encode methylated bases as complex misincorporation, as is seen in SHAPE MaP experiments, we utilized a previously published dataset where *E. coli* 23 s rRNA were reverse transcribed by SSII and Marathon RT (*Mitchell et al., 2023*). We downloaded the raw data from GSE225383 and combined all control samples for SSII and Marathon RT. We then aligned the reads to *E. coli* 23 s rRNA sequences with ShapeMapperV2.2 with the following settings: `--star-aligner --random-primer-len 9 --output-aligned-reads`. The aligned.sam file was then converted into a.bam file and imported into integrated genome viewer (IGV). The misincorporation rate was then recorded for each position relative to the modified bases G745 and U1915. The results show a clear increase in misincorporation rate for the first position after the modified nucleotide and this effect is more pronounced for SSII as compared with Martahon RT. This suggests that methylated bases may produce more complex misincorporation types rather than simple mismatches, particularly when using SSII. It is important to note that 2 bases beyond the site of modification, the misincorporation rate falls back to baseline.

We next applied the same analysis procedure to determine the optimal separation distance for BASH MaP experiments (*Figure 7—figure supplement 1a–b*). We utilized our own HeLa 18 s rRNA dataset where we depurinated $m^7G1638$ into an abasic site and reverse transcribed the rRNA with SSII and Marathon RT. We similarly converted the aligned.sam file into a.bam file and utilized IGV to count the misincorporation around G1638. We saw a clear increase in misincorporation rate after the abasic site for SSII samples but no increase in misincorporation rate for Marathon RT (*Figure 7—figure supplement 1a–b*). The pattern of misincorporation induced by abasic sites was like those seen for methylated bases in *E. coli* 23 s rRNA. In both cases, an adduct site does not appear to induce increased misincorporation at distances greater than two nucleotides. Together, these data suggest that the optimal misincorporation separation distance for BASH MaP is 2 nucleotides.

For Spinach, pUG, and AKT2 experiments, sequencing data was aligned to a single fasta sequence with ShapeMapperV2.2 with the following settings: `--amplicon --dms --output-parsed --min-mutation-separation 2`. Deletions are ambiguously aligned and therefore ignored by invoking the --dms option.

## Misincorporation signature analysis

Misincorporation signatures were calculated by aligning the data with ShapeMapperV2.2 with the following settings: `--amplicon --output-counted-mutations --min-mutation-separation 0`. Counts of misincorporation types for each position within an RNA are tabulated by the `--output-counted-mutations field`. A minimum misincorporation separation distance of 0 was chosen to capture the full range of misincorporation types generated in BASH MaP and DMS MaP experiments.

## Receiver operator curve generation

Bases in Spinach were annotated as either base-paired or single stranded based on the secondary structure provided in *Figure 1—figure supplement 1c*. Misincorporation rates were derived by probing Spinach in Bicine pH 8.3 buffer which promotes modification of $m^3U$ and $m^1G$. Although $m^1G$ reaction is promoted, $m^7G$ formation is favored by an order of magnitude. Misincorporation rates were calculated in ShapeMapperV2.2 with the follow settings: `--amplicon --dms --min-mutation-separation 2`. The crystal structure 4TS2 was utilized and all isolated base pairs and non-canonical base pairs in which the Watson-Crick face was not hydrogen bonded were designated as single stranded. This mainly affected annotation of A bases in the J1-2 region. ROC curves were generated with the python script ROC_curve_DAGGER.py and graphed with PRISM GraphPad.

## Correlated misincorporation analysis

ShapeMapper alignment with `--output-parsed produces` a.mut file which encodes the misincorporations within a sequencing read as a bitvector where 0 represents no misincorporation and 1 represents a misincorporation. This.mut file was then input into the program RingMapper which identifies positions with statistically high rates of co-occurring misincorporations (*Homan et al., 2014*). The output of RingMapper was processed with the custom python script ring_pair_to_heatmap.py which filters all negative correlations and creates a square matrix which was then visualized in Prism GraphPad.

## Network analysis of bases with correlated misincorporations

RingMapper was run with default parameters and the output file was filtered to include only positive correlations between two G nucleotides. A second filter removed all correlations with Z-scores less than 2.0. Networks of structurally related nucleotides were visualized in Cytoscape with unique G bases represented as nodes and positive correlations represented as edges.

## AKT2 G-quadruplex conformational analysis

The sequence GGGUGGGGAGGGGUGGGGUUGGUUCGGGUGGGUGAGGGU, corresponding to the putative G-rich sequence in the *AKT2* 3'UTR, was input to GQRS Mapper with the following search parameters: Max length = 45, Min G-Group size = 3, Loop size: from 0 to 36. All predicted G-quadruplex conformations with overlaps were then viewed and saved as an.html file before undergoing processing with the python script QGRS_to_average_mutation_rate.py. The output excel file was then filtered by average misincorporation rate and the ten lowest conformations were plotted in Prism GraphPad where lowercase 'g' characters indicate G's engaged in a G-quadruplex.

## M2 BASH MaP

Spinach and polyUG T7 templates were randomly mutagenized by performing 24 rounds of error-prone PCR as described previously (*Cheng et al., 2017*). Mutagenized Spinach and polyUG were then in vitro transcribed, folded and DMS modified as described prior with minor modifications. $MgCl_2$ was added to a final concentration of 5 mM and Spinach was probed in the presence of 5 µM DFHBI-1T. polyUG RNA was probed in the presence of 2 µM NMM.

For M2 Spinach and pUG experiments, sequencing data was aligned to a single fasta sequence with ShapeMapperV2.2 with the following settings: `--amplicon --dms --output-parsed --min-mutation-separation 2`. The output parsed mutation file was then processed with the custom script mut_to_simple.py which retains only sequencing reads that cover the entire length of the RNA. Then the output.simple file was used to calculate a unique mutational profile for each point mutant with the script simple_to_M2_map.py. This script produces a square matrix with length equal to the length of the RNA and displays how often two mutations co-occur on the same sequencing read.

To better visualize changes in nucleotide reactivity given a mutation at a certain position, Z-score normalization was performed with the script M2_Map_to_Zscores.py. The normalized matrix was then visualized in Prism GraphPad as a heatmap.

## Minimum free energy RNA secondary structure modeling

The baseline secondary structure model of Spinach was generated in mFold with default parameters. Population average DMS guided RNA secondary structure modeling was performed as described previously in the DANCE pipeline (*Olson et al., 2022*). Briefly, the python script DanceMapper.py was run on parsed mutation output files with the following settings for deriving population average data: `--fit --maxc 1`. Then, the population average reactivities were converted to free energy restraints for the RNA folding program RNAstructure through the foldClusters.py script. When processing BASH MaP data, the additional command --nog was used with foldClusters.py to ignore the reactivity data for G nucleotides. The resulting RNA secondary structures were visualized in RNA2D drawer. Mapping of per nucleotide DMS reactivity onto RNA2D drawer secondary models was performed through the custom python script color_reactivities_RNA2D_drawer.py. Conformation specific RNA secondary structure modeling was performed by increasing the maximum allowed conformations from 1 to 5 with --maxc 5.

## Single-molecule probabilistic RNA secondary structure modeling

A custom pipeline was developed to apply the DaVinci structural analysis method to BASH MaP (*Yang et al., 2022*). This analysis method utilizes misincorporations to generate a folding constraint for each sequencing read. Folding constraints are then passed to ContraFold which folds each sequencing read as a unique RNA molecule. The complete analysis pipeline is as follows:

First, sequencing data is processed through ShapeMapper as described above with `--amplicon --dms --output-parsed --min-mutation-separation` 2 options enabled. The resulting.mut file is then filtered and converted to a.simple file with mut_to_simple.py. For DMS MaP experiments, the resulting.simple file is converted into a.bit file by running the simple_to_bit.py python script. For BASH MaP experiments, the resulting.simple file is instead converted into a.bit file with the script updated_simple_to_bit_rG42.py. This script requires an input fasta file and sets all upper-case G positions to be considered for base pairing. In addition, the script constrains all lowercase letters to be single stranded in the.bit file. If a sequencing read harbors a misincorporations at the position of one of the lowercase letters, then all lowercase letters in that sequencing read are instead considered for base pairing. The resulting.bit file is then passed as an argument to the fold-contrafold-uniq-bits-vectors2.py script. ContraFold was run either with normal settings or with --noncomplementary enabled. Typically, the fold-contrafold-uniq-bits-vectors2.py script is stopped after roughly 10,000 RNA molecules have been folded to save on computation time. The forgi vector encodings of the RNA secondary structures generated by ContraFold are output in the file forgi-vect-ser2.txt. Dimensional reduction is performed on the forgi vectors text file with the python script run-pca-on-forgi-vectors.py which reduces the vector space to dimensions. Kmeans clustering is then performed on the 2-dimensional representation of RNA structures with the draw-kmeans-clusters.py script. Identification of a population average structure is done by choosing `--num_clusters 1`. Multiple conformational analysis is performed by varying the number of Kmeans clusters. The structure of the most representative of the Kmeans cluster is retrieved with the python script fetch_DaVinci_folded_copy.py and visualized with RNA 2D drawer.

## Nucleotide reactivity normalization

Mutation rates were normalized to reactivity values as previously described with the custom python script normalize_DAGGER_DANCE.py (*Mitchell et al., 2023*).

## Tertiary folding constraint determination and implementation (DAGGER)

G's likely to be engaged in tertiary interactions were identified by first selecting G's in the bottom quartile for misincorporation rate. In situations when reactivity data was available for multiple conformations, G's in separate conformations were treated as unique G's and all unique G's were pooled for reactivity normalization. From the bottom quartile of reactive G's, G's which displayed correlations

to other lowly reactive G's as identified by RingMapper were designated as engaged in a tertiary interaction.

The DaVinci analysis pipeline was modified to utilize tertiary constraints through setting tertiary G positions to lowercase G's in the fasta input file for generation of.bit files by updated_simple_to_bit_rG42.py. The modified DaVinci analysis pipeline is reffered to as DAGGER.

## Crystallographic structures

The reference Spinach crystal structure is 4TS2 (*Warner et al., 2014*). The reference polyUG crystal structure is 7MKT (*Roschdi et al., 2022*).

## In gel fluorescence assay

Spinach mutants were purchased from Twist and PCR amplified with the Structure cassette T7 forward and reverse primer. Fluorescence was assayed using a previously described in gel staining assay (*Filonov et al., 2015*). The protocol is as follows: 50 ng of RNA was denatured with 2 x formamide RNA loading buffer at 95 °C for 3 min and then loaded onto a 10% TBE – Urea PAGE gel. RNA was run at 280 V for 35 min and then the PAGE gel was washed in RNase-free water three times for 5 min each. Then the RNA was refolded and stained inside the gel by incubation with refolding buffer (10 µM DFHBI-1T, 40 mM HEPES pH 7.4, 100 mM KCl, 1 mM $MgCl_2$) for 15–20 min with shaking. The gel was imaged then washed in RNase-free water three times for 5 min each. The gel was then imaged in the SYBR gold channel to confirm all DFHBI-1T had been washed away. The gel was then stained with SYBR gold for 5 min and imaged again. Fluorescence was normalized to SYBR gold signal.

## Acknowledgements

We thank members of the Jaffrey lab for helpful comments and suggestions. We thank the Genomics Resources Core Facility at Weill Cornell for performing all DNA sequencing. This work was supported by NIH grants R35NS111631, RM1HG011563 and S10 OD030335 to SRJ, and NIH grant T32 GM141949-03 to MDO.

## Additional information

### Competing interests

Samie R Jaffrey: Co-founder of Lucerna Technologies and has equity in this company. Lucerna has licensed technology related to Spinach, Broccoli, and other RNA-fluorophore complexes. Founder of Chimerna Therapeutics and has equity in this company. The other author declares that no competing interests exist.

### Funding

| Funder | Grant reference number | Author |
|---|---|---|
| National Institute of Neurological Disorders and Stroke | R35NS111631 | Samie R Jaffrey |
| National Human Genome Research Institute | RM1HG011563 | Samie R Jaffrey |
| NIH Office of the Director | S10 OD030335 | Samie R Jaffrey |
| National Institute of General Medical Sciences | T32 GM141949-03 | Maxim Oleynikov |

The funders had no role in study design, data collection and interpretation, or the decision to submit the work for publication.

### Author contributions

Maxim Oleynikov, Conceptualization, Resources, Data curation, Software, Formal analysis, Funding acquisition, Validation, Investigation, Visualization, Methodology, Writing – original draft, Project

administration, Writing – review and editing; Samie R Jaffrey, Resources, Formal analysis, Supervision, Funding acquisition, Writing – original draft, Project administration, Writing – review and editing

**Author ORCIDs**
Maxim Oleynikov https://orcid.org/0000-0002-6892-9743
Samie R Jaffrey https://orcid.org/0000-0003-3615-6958

Reviewer #1 (Public review): https://doi.org/10.7554/eLife.98540.3.sa1
Reviewer #3 (Public review): https://doi.org/10.7554/eLife.98540.3.sa2
Author response https://doi.org/10.7554/eLife.98540.3.sa3

## Additional files

### Supplementary files

• MDAR checklist

• Supplementary file 1. Spinach G-quadruplex false positive rate estimation for various BASH MaP data processing approaches.

### Data availability

Raw and processed sequencing data have been deposited at GEO under the accession number GSE271825. All code used for analysis is available at GitHub, copy archived at *Jaffrey, 2024*.

The following dataset was generated:

| Author(s) | Year | Dataset title | Dataset URL | Database and Identifier |
|---|---|---|---|---|
| Oleynikov M, Jaffrey S | 2024 | RNA tertiary structure and conformational dynamics revealed by BASH MaP | https://www.ncbi.nlm.nih.gov/geo/query/acc.cgi?acc=GSE271825 | NCBI Gene Expression Omnibus, GSE271825 |

The following previously published dataset was used:

| Author(s) | Year | Dataset title | Dataset URL | Database and Identifier |
|---|---|---|---|---|
| Mitchell III D, Cotter J, Saleem I, Mustoe AM | 2023 | Enhanced DMS-MaP enables superior RNA structural analysis | https://www.ncbi.nlm.nih.gov/geo/query/acc.cgi?acc=GSE225383 | NCBI Gene Expression Omnibus, GSE225383 |

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
