## [Editor Report · eLife Assessment]

This **important** work substantially advances our understanding of RNA structure analysis by introducing an innovative method that extends DMS probing to include guanosine residues, thereby enhancing our ability to detect complex tertiary interactions. The evidence supporting the conclusions is **compelling**, with detailed analyses demonstrating the method's capacity to differentiate structural contexts and improve RNA structure predictions. This work will be of broad interest to RNA structural biology, biochemistry, and biophysics researchers.

---

## [Referee Report · Reviewer #1 (Public review)]

Summary:

DMS-MaP is a sequencing-based method for assessing RNA folding by detecting methyl adducts on unpaired A and C residues created by treatment with dimethylsulfate (DMS). DMS also creates methyl adducts on the N7 position of G, which could be sensitive to tertiary interactions with that atom, but N7-methyl adducts cannot be detected directly by sequencing. In this work, the authors adopt a previously developed method for converting N7-methyl-G to an abasic site to make it detectable by sequencing and then show that the ability of DMS to form an N7-methyl-G adduct is sensitive to RNA structural context. In particular, they look at the G-quadruplex structure motif, which is dense with N7-G interactions, is biologically important, and lacks conclusive methods for in-cell structural analysis.

Strengths:

- The authors clearly show that established methods for detecting N7-methyl-G adducts can be used to detect those adducts from DMS and that the formation of those adducts is sensitive to structural context, particularly G-quadruplexes.

- The authors assess the N7-methyl-G signal through a wide range of useful probing analyses, including standard folding, adduct correlations, mutate-and-map, and single-read clustering.

- The authors show encouraging preliminary results toward the detection of G-quadruplexes in cells using their method. Reliable detection of RNA G-quadruplexes in cells is a major limitation for the field and this result could lead to a significant advance.

- Overall, the work shows convincingly that N7-methyl-G adducts from DMS provide valuable structural information and that established data analyses can be adapted to incorporate the information.

Weaknesses:

- Most of the validation work is done on the spinach aptamer and it and polyUG RNA are the only RNAs tested that have a known 3D structure. Although it is a useful model for validating this method, it does not provide a comprehensive view of what results to expect across varied RNA structures.

- It's not clear from this work what the predictive power of BASH-MaP would be when trying to identify G-quadruplexes in RNA sequences of unknown structure. Although clusters of G's with low reactivity and correlated mutations seem to be a strong signal for G-quadruplexes, no effort was made to test a range of G-rich sequences that are known to form G-quadruplexes or not. Having this information would be critical for assessing the ability of BASH-MaP to identify G-quadruplexes in cells.

- Although the authors present interesting results from various types of analysis, the code currently available on Github lacks the documentation and examples necessary to be useful to the broader community.

- There are aspects of the DAGGER analysis that could limit its robustness or utility for different RNAs:

(1) Folding of the RNA based on individual reads does not represent single-molecule folding since each read contains only a small fraction of the possible adducts that could have formed on that molecule. As a result, each fold will largely be driven by the naive folding algorithm. The DANCE-MaP algorithm that was also used by the authors addresses this concern.

(2) G residues in a loop will have a different impact on RNA folding than those in a G-quadruplex. This difference could reduce the accuracy of CONTRAfold predictions when forcing G-quadruplex residues to be unpaired. That said, predicting secondary structure around G-quadruplexes is a challenge for folding algorithms.

(3) Incorporation of the G mutations requires prior knowledge of the RNA 3D structure, limiting the utility of the method to predicting alternative conformations in structures that are already well characterized.

---

## [Referee Report · Reviewer #3 (Public review)]

Summary:

In this study the authors aim to develop an experimental/computational pipeline to assess the modification status of an RNA following treatment with dimethylsulfate (DMS). Building upon the more common DMS Map method, which predominantly assesses the modification status of the Watson-Crick-Franklin face of A's and C's, the authors insert a chemical processing step in the workflow prior to deep sequencing that enables detection of methylation at the N7 position of guanosine residues. This approach, termed BASH MaP, provides a more complete assessment of the true modification status of an RNA following DMS treatment, and this new information provides a powerful set of constraints for assessing the secondary structure and conformational state of an RNA. In developing this work, the authors use Spinach as a model RNA. Spinach is a fluorogenic RNA that binds and activates the fluorescence of a small molecule ligand. Crystal structures of this RNA with ligand bound show that it contains a G-quadruplex motif. In applying BASH MaP to Spinach, the authors also perform the more standard DMS MaP for comparison. They show that the BASH MaP workflow appears to retain the information yielded by DMS MaP while providing new information about guanosine modifications. In Spinach, the G-quadruplex G's have the least reactive N7 positions, consistent with the engagement of N7 in hydrogen bonding interactions at G's involved in quadruplex formation. Moreover, because the inclusion of data corresponding to G increases the number of misincorporations per transcript, BASH MaP is more amenable to analysis of co-occurring misincorporations through statistical analysis, especially in combination with site-specific mutations. These co-occurring misincorporations provide information regarding what nucleotides are structurally coupled within an RNA conformation. By deploying a likelihood-ratio statistical test on BASH MaP data, the authors can identify Gs in G-quadruplexes, deconvolute G-G correlation networks, base-triple interactions and even stacking interactions. Further, the authors develop a pipeline to use the BASH MaP-derived G-modification data to assist in the prediction of RNA secondary structure and identify alternative conformations adopted by a particular RNA. This seems to help with the prediction of secondary structure for Spinach RNA.

Strengths:

The BASH Map procedure and downstream data analysis pipeline more fully identifies the complement of methylations to be identified from DMS treatment of RNA, thereby enriching the information content. This in turn allows for more robust computational/statistical analysis, which likely will lead to more accurate structure predictions. This seems to be the case for the Spinach RNA.

Weaknesses:

The authors demonstrate that their method can detect G-quadruplexes in Spinach and some other RNAs both in vitro and in cells. While application to other RNAs is beyond the scope of the current manuscript, the performance of BASH MaP and associated computational analysis in the context of other RNAs remains to be determined.

---

## [Author Response]

The following is the authors’ response to the original reviews.

**Public Reviews:**

**Reviewer #1 (Public Review):**
Summary:DMS-MaP is a sequencing-based method for assessing RNA folding by detecting methyl adducts on unpaired A and C residues created by treatment with dimethylsulfate (DMS). DMS also creates methyl adducts on the N7 position of G, which could be sensitive to tertiary interactions with that atom, but N7-methyl adducts cannot be detected directly by sequencing. In this work, the authors adopt a previously developed method for converting N7-methyl-G to an abasic site to make it detectable by sequencing and then show that the ability of DMS to form an N7-methyl-G adduct is sensitive to RNA structural context. In particular, they look at the G-quadruplex structure motif, which is dense with N7-G interactions, is biologically important, and lacks conclusive methods for in-cell structural analysis.Strengths:- The authors clearly show that established methods for detecting N7-methyl-G adducts can be used to detect those adducts from DMS and that the formation of those adducts is sensitive to structural context, particularly G-quadruplexes.- The authors assess the N7-methyl-G signal through a wide range of useful probing analyses, including standard folding, adduct correlations, mutate-and-map, and single-read clustering.- The authors show encouraging preliminary results toward the detection of G-quadruplexes in cells using their method. Reliable detection of RNA G-quadruplexes in cells is a major limitation for the field and this result could lead to a significant advance.- Overall, the work shows convincingly that N7-methyl-G adducts from DMS provide valuable structural information and that established data analyses can be adapted to incorporate the information.

We thank the reviewer for their time and appreciate the reviewer for their positive assessment as well as for their suggestions which we have addressed below.

Weaknesses:- Most of the validation work is done on the spinach aptamer and it is the only RNA tested that has a known 3D structure. Although it is a useful model for validating this method, it does not provide a comprehensive view of what results to expect across varied RNA structures.

Thank you for your insightful comments. We agree that a more comprehensive view of BASH MaP involves probing a larger variety of RNAs with known 3D-structures beyond Spinach and the poly-UG RNA. Although outside the scope of this publication, more work is needed to reveal the determinants of N7G reactivity to DMS.

- It's not clear from this work what the predictive power of BASH-MaP would be when trying to identify G-quadruplexes in RNA sequences of unknown structure. Although clusters of G's with low reactivity and correlated mutations seem to be a strong signal for G-quadruplexes, no effort was made to test a range of G-rich sequences that are known to form G-quadruplexes or not. Having this information would be critical for assessing the ability of BASH-MaP to identify G-quadruplexes in cells.- Although the authors present interesting results from various types of analysis, they do not appear to have developed a mature analysis pipeline for the community to use. I would be inclined to develop my own pipeline if I were to use this method.

Thank you for your suggestion. We have more clearly annotated the python scripts and GitHub repository which contain all custom scripts used for analyzing BASH MaP data. These changes will enable researchers to more easily utilize our developed pipelines.

- There are various aspects of the DAGGER analysis that don't make sense to me:(1) Folding of the RNA based on individual reads does not represent single-molecule folding since each read contains only a small fraction of the possible adducts that could have formed on that molecule. As a result, each fold will largely be driven by the naive folding algorithm. I recommend a method like DREEM that clusters reads into profiles representing different conformations.(2) How reliable is it to force open clusters of low-reactivity G's across RNA's that don't already have known G-quadruplexes?(3) By forcing a G-quadruplex open it will be treated as a loop by the folding algorithm, so the energetics won't be accurate.(4) It's not clear how signals on "normal" G's are treated. In Figure 5C some are wiped to 0 but others are kept as 1.

Thank you for your keen observations regarding the conceptual frameworks utilized in DAGGER. We have included a complimentary analysis to DAGGER utilizing Spinach BASH MaP data with DANCE, an algorithm which shares an underlying architecture with DREEM, and found that DANCE analysis gave similar results to those found with DAGGER. However, we have not benchmarked DAGGER’s performance on a range of RNAs and compared the results with expectation-maximization algorithms like DREEM and DANCE.

To minimize the effects of artificially creating loops with tertiary folding constraints, we utilized the RNA folding algorithm CONTRAfold which relies less on direct energetic calculations than other commonly used RNA folding algorithms such as RNAstructure.

We have updated the main text to more clearly indicate how DAGGER handles signals at G’s in a range of conditions. The main text now better clarifies the specific logic used for determining which G’s contain either a 0 or a 1 in the bitvector encoding used in DAGGER analysis.

**Reviewer #2 (Public Review):**
Summary:The manuscript introduces BASH MaP and DAGGER, innovative tools for analyzing RNA tertiary structures, specifically focusing on the G-quadruplexes. Traditional methods have struggled to detect and analyze these structures due to their reliance on interactions on the Hoogsteen face of guanine, which are not readily observable through conventional probing that targets Watson-Crick interactions. BASH MaP employs dimethyl sulfate and potassium borohydride to enhance the detection of N7-methylguanosine by converting it into an abasic site, thereby enabling its identification through misincorporation during reverse transcription. This method provides higher precision in identifying G-quadruplexes and offers deeper insights into RNA's structural dynamics and alternative conformations in both vitro and cellular contexts. Overall, the study is well-executed, demonstrating robust signal detection of N7-Gs with some compelling positive controls, thorough analysis, and beautifully presented figures.Strengths:The manuscript introduces a new method to detect G-quadruplexes (G-qs) that simplifies and potentially enhances the robustness and quantification compared to previous methods relying on reverse transcription truncations. The authors provide a strong positive control, demonstrating a 70% misincorporation at endogenous N7-G within the 18S rRNA, which illustrates BASH MaP's high signal-to-noise ratio. The data concerning the detection of positive control G-qs is particularly compelling.Weaknesses:Figure 3E shows considerable variability in the correlations among guanosines, suggesting that the methods may struggle with specificity in determining guanosine participation within and between different quadruplexes. There is no estimation of the methods false positive discovery rate.

Thank you for your positive assessment and for your time to come up with suggestions to improve this publication. We have addressed your specific comments in the “Recommendations For The Authors” section below.

**Reviewer #3 (Public Review):**
Summary:In this study, the authors aim to develop an experimental/computational pipeline to assess the modification status of an RNA following treatment with dimethylsulfate (DMS). Building upon the more common DMS Map method, which predominantly assesses the modification status of the Watson-Crick-Franklin face of A's and C's, the authors insert a chemical processing step in the workflow prior to deep sequencing that enables detection of methylation at the N7 position of guanosine residues. This approach, termed BASH MaP, provides a more complete assessment of the true modification status of an RNA following DMS treatment and this new information provides a powerful set of constraints for assessing the secondary structure and conformational state of an RNA. In developing this work, the authors use Spinach as a model RNA. Spinach is a fluorogenic RNA that binds and activates the fluorescence of a small molecule ligand. Crystal structures of this RNA with ligand bound show that it contains a G-quadruplex motif. In applying BASH MaP to Spinach, the authors also perform the more standard DMS MaP for comparison. They show that the BASH MaP workflow appears to retain the information yielded by DMS MaP while providing new information about guanosine modifications. In Spinach, the G-quadruplex G's have the least reactive N7 positions, consistent with the engagement of N7 in hydrogen bonding interactions at G's involved in quadruplex formation. Moreover, because the inclusion of data corresponding to G increases the number of misincorporations per transcript, BASH MaP is more amenable to analysis of co-occurring misincorporations through statistical analysis, especially in combination with site-specific mutations. These co-occurring misincorporations provide information regarding what nucleotides are structurally coupled within an RNA conformation. By deploying a likelihood-ratio statistical test on BASH MaP data, the authors can identify Gs in G-quadruplexes, deconvolute G-G correlation networks, base-triple interactions and even stacking interactions. Further, the authors develop a pipeline to use the BASH MaP-derived G-modification data to assist in the prediction of RNA secondary structure and identify alternative conformations adopted by a particular RNA. This seems to help with the prediction of secondary structure for Spinach RNA.Strengths:The BASH Map procedure and downstream data analysis pipeline more fully identify the complement of methylations to be identified from the DMS treatment of RNA, thereby enriching the information content. This in turn allows for more robust computational/statistical analysis, which likely will lead to more accurate structure predictions. This seems to be the case for the Spinach RNA.Weaknesses:The authors demonstrate that their method can detect G-quadruplexes in Spinach and some other RNAs both in vitro and in cells. However, the performance of BASH MaP and associated computational analysis in the context of other RNAs remains to be determined.

We thank the reviewer for their time spent analyzing this manuscript, for their positive assessment and for their suggestions on improving this publication. We have addressed your specific comments in the “Recommendations For The Authors” section below.

**Recommendations for the authors:**

**Reviewer #1 (Recommendations For The Authors):**
Although the text is clear and coherent, the overall flow of the manuscript comes across as "here's a bunch of stuff I tried." Maybe you're looking to get this out quickly, but it would have been much more impactful (and enjoyable to read) a description of a more polished final product.

Thank you for your highlighting the strengths and weaknesses of this manuscript. We have changed parts of the main text to enhance the overall flow of the manuscript and increase reader enjoyability.

**Reviewer #2 (Recommendations For The Authors):**
I have only a few comments:Major:(1) Analysis of Guanosine Correlations in Figure 3E: In Figure 3E, there is a lot of variability in the correlations among guanosines. For example, G46 shows a strong correlation with G93 (within the same quadruplex) but also correlates with G91, G95 (in different quadruplexes), and G97 (not part of any quadruplex as per the model in Figure 3C). Contrarily, G86 exhibits weak correlations, and G50 along with G89 shows no significant correlations. These findings imply that BASH MaP followed by RING MaP analysis struggles to accurately distinguish between guanosines within the same or different quadruplexes in Spinach. Perhaps there are some opportunities to enhance the specificity in determining guanosine participation within quadruples, a great point for the authors to discuss.

Thank you for your comments and careful analysis of the pattern of correlations produced by BASH MaP. We agree that BASH MaP followed by RING MaP analysis is unable to unambiguously distinguish between guanosines within the same or different quadruplex layers. This finding was a surprise as we initially assumed that quadruplex layers would behave in a manner like Watson-Crick base pairs and produce specific signals in the corresponding RING MaP heatmaps. We suspect that this may be due to mutations in specific G’s being associated with altered conformations which allow other G’s to form different interactions that affect DMS reactivity. This may be unique to the highly complex structure in Spinach. However, we think BASH-MaP clearly provides signals that point to key residues within the G-quadruplex, even if it does not clearly identify all of them.

This idea is supported by experiments described in Figure 4, which show that mutation of a single guanosine residue causes a complete breakdown of the hydrogen-bonding network throughout all quadruplex layers. Additionally, DMS methylation of an N7G in a quadruplex is likely to disrupt base stacking interactions in and around the quadruplex. The compounding effects of a dynamic G-quadruplex and DMS-induced changes to local base stacking properties explains both the strong correlations with G97, which is base-stacked with the quadruplex, and the inability to specifically identify the guanosines which comprise specific quadruplex quartets. We have further emphasized this point in an updated discussion section.

(2) Potential Consolidation of Figures 3 and 4: Figure 4 appears quite similar to Figure 3 but employs M2-seq instead of relying on spontaneous mutations. It might be beneficial to merge these figures to demonstrate that M2-seq can more effectively identify correlations between guanosines in quadruplexes.

We agree that Figures 3 and 4 appear quite similar but there is an important distinction to be made between RING MaP and M2-seq analysis. We suspect that the mechanism causing correlations between guanosines in quadruplexes for RING MaP as “RNA breathing” in contrast to the spontaneous T7 RNA polymerase-induced mutation model proposed in Cheng et al. PNAS 2017, https://doi.org/10.1073/pnas.1619897114. To determine whether correlations between guanosines in Spinach BASH MaP experiments rely on spontaneous mutations, we compared the fraction of reads containing misincorporations at pairs of quadruplex guanosines over a range of DMS concentrations. The spontaneous mutation model predicts a linear dependence between quadruplex guanosine signals and DMS dose while an “RNA breathing” or double-DMS hit model predicts a quadratic dependence on DMS dose (Cheng et al. PNAS 2017, https://doi.org/10.1073/pnas.1619897114). Our data may support a quadratic dependence on DMS dose for multiple pairs of G-quadruplex guanosines, while they demonstrate a linear dependence between helical G’s (Supplementary Data Fig. 9). Together, these data suggest that BASH MaP followed by RING MaP analysis detects double-DMS modification events for pairs of quadruplex guanosines. Therefore, BASH MaP and RING MaP analysis provide a complimentary approach to M2 BASH MaP and reveal guanosine correlations in contexts where pre-installed mutations are incompatible such as the study of endogenously expressed RNAs.

(3) Estimation of False Positive Rates: An estimation of the false positive rate for G-quadruplex identification would be invaluable. Since identification currently depends on the absence of DMS modification, it's important to consider how other factors like solvent inaccessibility or library generation might affect the detection and be misinterpreted as G-quadruplexes. Although this could be a subject of future work, some discussion by the authors would enhance the manuscript.

We have added a table summarizing sensitivity, positive predictive value, and false positive rate for different G-quadruplex identification schemes. See Supplementary Table 1.

Minor:(4) Line 273 Reference Correction: Please adjust the reference in line 273 to accurately reflect that the G-quadruplex experiments compare potassium with lithium, not sodium.

In cellulo G-quadruplex reverse transcriptase (RT) stop assays as described by Guo and Bartel (https://www.science.org/doi/10.1126/science.aaf537) compared RT stops between DMS treated mRNA refolded in potassium and sodium buffers. We have clarified in the text that traditionally, G-quadruplex RT stop assays compare potassium with lithium.

(5) Consistency in Figure 1 (Panels F and G): Aligning BASH MaP (170 mM DMS) as the y-axis in both panels F and G would visually align the data points and enhance the graphical coherence across these panels.

Thank you for noticing the subtleties in our data presentation and for the suggestion on how to improve our graphical coherence across panels. We specifically choose not to align BASH MaP (170 mM DMS) as the y-axis for panels F and G because we did not want the reader to mistakenly assume that the data for BASH MaP (170 mM DMS) presented in panels F and G is the same data. In panel F, BASH MaP was performed under standard DMS probing buffer conditions which utilized a pH 7.5 bicine buffer. The purpose of panel F is to show the reproducibility of BASH MaP under various DMS concentrations. In panel G, BASH MaP was performed under DMS probing buffer conditions which promote the formation of m3U using a pH 8.3 bicine buffer. The purpose of panel G is to show that the borohydride treatment and depurination steps in BASH MaP do not react with DMS-derived m1A, m3C, and m3U in a manner which prevents their measurement through cDNA misincorporation. Together, these experimental differences cause the data points for BASH MaP (170 mM DMS) to vary between panels F and G which would lead to more confusion for the reader and detract from the intended message we are trying to convey through panels F and G.

(6) Statistical Detail in Figure 1E: Incorporating a confidence interval or a P-value in Figure 1E would enrich the statistical depth and provide readers with a clearer understanding of the data's significance.

Thank you for the suggestion of including a p-value in Figure 1E to provide the readers with a clearer understanding of the data’s significance. The effect of combining DMS treatment and borohydride reduction on the misincorporation rate of G’s in Spinach is so dramatic that the raw data sufficiently provides the readers a clear understanding of its significance.

(7) Reevaluation of Figure 2B: Considering the small number of Gs in single-stranded regions and base triples, it might be more informative to move Figure 2B to supplementary information. Focusing on Figure 2C, which consolidates non-quadruplex categories, could provide more impactful insights.

Thank you for your suggestion. It is important to initially provide an overall characterization of N7G DMS reactivity for G’s in a variety of structural contexts before more specifically looking at G-quadruplexes. Panel B is an important part of figure 2 for the following two reasons:

First, a reader’s first question upon seeing the N7G chemical reactivity for Spinach as showed in Figure 2A is likely to ask whether base-paired G’s and single-stranded G’s have similar or different DMS reactivities. Figure 2, panel B shows that generally, single-stranded G’s appear to have higher DMS reactivity than base-paired G’s except for 2 G’s which display hyper-reactivity. The basis for this hyper-reactivity is addressed in Figure 4.

Second, panel B highlights the wide range in N7G DMS reactivities. Since the G-quadruplex G’s display a dramatically lower DMS reactivity as compared to single-stranded G’s and hyper-reactive base-paired G’s, the dynamic range of DMS reactivities was difficult to capture in a single panel. Panel C does not convey these dynamics appropriately as a stand-alone figure.

(8) Enhancements to Figure 2G: Improving the visibility of mutation rates in this figure would help. Suggestions include coloring bars by nucleotide type for intuitive visual comparison and adjusting the y-axis to a logarithmic scale to better represent near-zero mutation rates. Additionally, employing histograms or box plots could directly compare DMS reactivities and provide a clearer analysis.

Thank you for your suggestions on enhancing the presentation of BASH MaP applied to an mRNA. The main purpose of figure 2G was to validate whether BASH MaP could detect G’s engaged in a G-quadruplex in a cell. In-cell G-quadruplex folding measurements as performed by Guo and Bartel (https://www.science.org/doi/10.1126/science.aaf537) only identified a few G-quadruplexes which were folded and only the 3’ end of the G-quadruplex was detected. We therefore reasoned that the 3’ most G’s of these select set of G-quadruplexes were the only validated G’s engaged in a G-quadruplex in cells. In the instance of the AKT2 mRNA, Guo and Bartel found that 4 G’s appeared to be folded in a G-quadruplex in cells (Supplementary figure 2E). These G’s are indicated at the bottom of the plot with black bars and the label “In-cell G-quadruplex guanosines”. Therefore, we hypothesized that these G’s would display low DMS reactivity with BASH MaP while other G’s in the AKT2 mRNA would display higher chemical reactivities. We followed a standard convention in displaying chemical reactivities used extensively in the field where black bars indicate low reactivity, yellow bars indicate moderate reactivity, and red bars indicate high reactivity. The data in Fig 2G directly supports Guo and Bartel’s prediction of an in-cell folded G-quadruplex in the AKT2 mRNA because the 4 G’s predicted to be engaged in a G-quadruplex all displayed near zero DMS reactivities.

We agree that adjusting the y-axis to a logarithmic scale would better represent near-zero mutations rates. However, the purpose of figure 2G is not to compare all positions with near-zero mutation rates. Instead, our use of standard conventions in displaying chemical reactivities is sufficient for the purpose of displaying BASH MaP’s ability to validate in-cell G-quadruplex G’s.

Later in the paper, we go a step further and create a better criterion than simple N7G DMS reactivity for identifying G’s engaged in a G-quadruplex. For further analysis of G’s with near zero DMS reactivities, see Figure 3 and Supplementary figure 4 which utilizes RING Mapper to identify lowly-reactive G’s which produce co-occurring misincorporations.

(9) Scale Consistency in Figure 3: Ensuring that the correlation scales are uniform across Panels A, B, D, and E would facilitate easier comparison of the data, enhancing the overall coherence of the findings. Using raw correlation values could also improve clarity and interpretation.

Thank you for the suggestions to facilitate easier comparisons of data in Figure 3. We have ensured the correlation scales are uniform across panels A, B, D, and E to enhance the coherence of these findings. We initially visualized the data in Figure 3 by plotting raw correlation values, but we found these values differed between DMS MaP and BASH MaP datasets, likely because of the low-level background mutations introduced by the borohydride reduction step of BASH (see Supplementary figure 3A). However, performing a global normalization of correlation strength values computed by RING mapper enabled clear comparisons between DMS MaP and BASH MaP RING heatmaps and revealed structural domains consistent with the crystal structure of Spinach.

(10) Correction on Line 506: Please update the reference to M2 BASH MaP for accuracy.

Thank you. We have updated the main text to incorporate this comment.

**Reviewer #3 (Recommendations For The Authors):**
The paper describes multiple applications and multiple methods of analysis of the BASH Map data, which collectively make the manuscript more difficult to follow. The manuscript would become more readable and user-friendly if there were some overview figures to describe the sequencing pipeline and the various computational workflows that the BASH MaP data are fed into (e.g. RING Mapper, DAGGER, M2 BASH MaP, Co-occurring Misincorporations, Secondary Structure Prediction). One or more summary schemes that provide an overview would strongly assist with the clarity and overall content of the paper.

Thank you for your suggestions. We have incorporated a summary scheme of the various computational workflows and their use cases in Fig 7.

Line 165. Here, misincorporation rates for all four nucleotides are discussed, but m3U is not mentioned until from the following paragraph. It would be appropriate and clearer to mention this sooner.

Thank you for your suggestion. We have restructured this section to introduce the DMS modification m3U in an earlier paragraph to increase clarity for readers.

Line 506: spelling of DAGGER.

Thank you. We have updated the main text to incorporate this comment.

Line 645: I found this paragraph difficult to follow, especially the line starting 649. I thought the logic was to exclude G's involved in tertiary interactions from base-paring in the secondary structure prediction. Some clarification would be helpful.

Thank you for your comments. We have restructured the paragraph to emphasize that DAGGER only applies tertiary folding constraints to sequencing reads without misincorporations at G’s engaged in tertiary interactions. We reasoned that sequencing reads with a misincorporation at a G engaged in a tertiary interaction likely come from an RNA molecule which is in an alternative tertiary conformational state. In this specific circumstance, a tertiary folding constraint may impose incorrect restrictions on the folding of RNA molecules due to distinct tertiary conformations.

Line 817. "Ability to".

Thank you. We have updated the main text to incorporate this comment.

Figure 6F. Mistake in the axis description.

Thank you. We have updated the main text to incorporate this comment.

Consider combining the paragraphs at lines 850 and 903.

Thank you for the suggestion. We rearranged paragraphs in the discussion to improve clarity.

Line 1546. The final conc of DMS would be nice to see here.

Thank you. We have updated the main text to incorporate this comment.